# LTL-Constrained Policy Optimization with Cycle Experience Replay

**Ameesh Shah**[*]  *ameesh@berkeley.edu*
*UC Berkeley*

**Cameron Voloshin**[*]  *cavoloshin@lat.ai*
*Latitude AI*

**Chenxi Yang**  *cxyang19@utexas.edu*
*UT Austin*

**Abhinav Verma**  *verma@psu.edu*
*Penn State University*

**Swarat Chaudhuri**  *swarat@cs.utexas.edu*
*UT Austin*

**Sanjit A. Seshia**  *sseshia@eecs.berkeley.edu*
*UC Berkeley*

**Reviewed on OpenReview:** *https://openreview.net/forum?id=gxUp2d4JTw*

## Abstract

Linear Temporal Logic (LTL) offers a precise means for constraining the behavior of reinforcement learning agents. However, in many settings where both satisfaction and optimality conditions are present, LTL is insufficient to capture both. Instead, LTL-constrained policy optimization, where the goal is to optimize a scalar reward under LTL constraints, is needed. This constrained optimization problem proves difficult in deep Reinforcement Learning (DRL) settings, where learned policies often ignore the LTL constraint due to the sparse nature of LTL satisfaction. To alleviate the sparsity issue, we introduce Cycle Experience Replay (CyclER), a novel reward shaping technique that exploits the underlying structure of the LTL constraint to guide a policy towards satisfaction by encouraging partial behaviors compliant with the constraint. We provide a theoretical guarantee that optimizing CyclER will achieve policies that satisfy the LTL constraint with near-optimal probability. We evaluate CyclER in three continuous control domains. Our experimental results show that optimizing CyclER in tandem with the existing scalar reward outperforms existing reward-shaping methods at finding performant LTL-satisfying policies.

## 1 Introduction

Significant research effort has explored *Linear Temporal Logic* (LTL) as an alternative to Markovian reward functions for specifying objectives for reinforcement learning (RL) agents (Sadigh et al., 2014; Hasanbeig et al., 2018; Camacho et al., 2019; Wang et al., 2020; Vaezipoor et al., 2021; Alur et al., 2022; De Giacomo et al., 2020; Voloshin et al., 2023). LTL provides a flexible language for defining objectives, or *specifications*, that are often not reducible to scalar Markovian rewards (Abel et al., 2021). Unlike typical reward functions, objectives defined in LTL are composable, easily transferred across environments, and offer a precise notion of satisfaction.

LTL specifications and Markovian reward functions have been used separately in a variety of RL settings, but few works consider both rewards *and* specifications in the same setting (Voloshin et al., 2022). The combination of the two is important: an LTL specification can define the meaning of achieving a task, and a reward function can be optimized to find the best way of achieving that task. For example, in robot motion planning, an LTL specification can describe the waypoints a robot should reach and obstacles it should avoid, and a reward function can optimize for factors like energy consumption, stability of motion, and so forth.

This work considers the problem setting of RL-based reward optimization under an LTL constraint. This constrained learning problem can be naturally formulated in an unconstrained form through a Lagrange-style relaxation (Le et al., 2019; Achiam et al., 2017) where the LTL constraint is represented by a *proxy* reward function. Some versions of this proxy have been introduced in prior literature (Hasanbeig et al., 2020; Voloshin et al., 2023; Camacho et al., 2019). However, these proxy rewards are sparse and difficult to optimize. As a result, learned policies in practice often end up ignoring the LTL constraint entirely and focus only on optimizing the reward function. A few existing works attempt to circumvent this sparsity issue by considering alternative temporal logics that provide a denser reward signal (Krasowski et al., 2023; Lavaei et al., 2020; Gundana & Kress-Gazit, 2021) or by limiting their approach to discrete state spaces that can be solved exactly (Voloshin et al., 2022).

In this paper, we introducing a novel reward-shaping proxy for LTL called *Cycle Experience Replay* (CyclER) that alleviates the reward-sparsity issue of previous LTL proxy rewards, allowing us to scale to continuous state and action spaces with an objective readily optimizable by deep RL (DRL) approaches. We briefly summarize the CyclER approach as follows. Given the (known) automaton structure representing an LTL objective, CyclER computes all possible infinite paths, or cycles, through the automaton that define accepting behaviors for our task. Then, when an agent collects a finite episode of experience in the world, CyclER counterfactually reasons over that experience by considering how much progress it made through each cycle. The cycle that the agent progressed through the furthest is then used to shape LTL reward. We provide a visualization of this approach in Figure 1.

Under certain assumptions, CyclER maintains theoretical guarantees on LTL optimality that are competitive with state-of-the-art LTL proxy rewards (Voloshin et al., 2023). A key advantage of CyclER is how it readily incorporates quantitative semantics (QS), a popular technique for reward design in temporal logic that has yet to be extended to infinite-horizon LTL tasks. Our empirical results demonstrate CyclER's effectiveness during policy learning.

In summary, this paper makes the following contributions. We present a problem formulation for LTL-constrained policy optimization in the presence of continuous spaces and function approximators for use in deep RL.

We propose a technique for this problem setting, CyclER, that alleviates the proxy reward sparsity issue and provides guarantees that ensure approximate optimality of LTL satisfaction.

Third, we introduce a new way to leverage quantitative semantics in LTL reward shaping. Lastly, we present promising experimental results using CyclER in LTL-constrained optimization settings, outperforming existing approaches.

## 2 Problem Setting

### 2.1 Preliminaries

An *atomic proposition* is a variable that takes on a Boolean truth value. We define an *alphabet* $\Sigma$ as all possible combinations over a finite set of atomic propositions (AP); that is, $\Sigma = 2^{\text{AP}}$. For example, if $\text{AP} = \{a, b\}$, then $\Sigma = \{\{a, b\}, \{b\}, \{a\}, \{\}\}$. We will refer to individual combinations of atomic propositions, or predicates, in $\Sigma$ as $\nu$.

**Labeled MDPs** We formulate our environment as a labelled Markov Decision Process $\mathcal{M} = (\mathcal{S}, \mathcal{A}, T^{\mathcal{M}}, d_0, \gamma, r, L^{\mathcal{M}})$, containing a state space $\mathcal{S}$, an action space $\mathcal{A}$, an *unknown* transition function, $T^{\mathcal{M}} : \mathcal{S} \times \mathcal{A} \to \Delta(\mathcal{S})$, an initial state distribution $d_0 \in \Delta(\mathcal{S})$, a discount factor $0 < \gamma < 1$, a reward function

Figure 1: Left: The FlatWorld MDP and an example trajectory. Right: An LDBA for the LTL formula $\varphi = G(F(r)\&F(g)\&F(y))\&G(\neg b)$ (some edges omitted for readability); the accepting state 0 is coded in green. The CyclER method considers all accepting paths within an LDBA and selects the most reward-ful path for the trajectory to shape LTL reward. Unlike previous approaches (Camacho et al., 2019; Voloshin et al., 2023), CyclER offers dense reward, even without visiting the accepting state.

$r : \mathcal{S} \times \mathcal{A} \to [R_{\min}, R_{\max}]$, and a labelling function $L^{\mathcal{M}} : \mathcal{S} \to \Sigma$. The labelling function returns which atomic propositions in our set AP are true for a given MDP state.

As a running example, consider the "FlatWorld" MDP in Figure 1 (left), where an agent (represented by the green dot) can move around the environment and visit different colored regions of interest. The agent is given a reward of 1 every time it visits one of the small purple regions. Our alphabet in FlatWorld is defined as $AP = \{r, g, b, y\}$, corresponding to whether the agent is in the red, green, blue, or yellow region at any point in time.

**Linear Temporal Logic (LTL)** Linear Temporal Logic (Pnueli, 1977) is a specification language that composes atomic propositions with logical and temporal operators to precisely define tasks. We use the symbol $\varphi$ to refer to an LTL task specification, also called an LTL formula.

In LTL, specifications are constructed using both logical connectives: not ($\neg$), and ($\&$), and implies ($\to$); and temporal operators: next ($X$), repeatedly/always/globally ($G$), eventually ($F$), and until ($U$). For more detail on the exact semantics of LTL operators, see Baier & Katoen (2008).

As an example, let us demonstrate task specifications in the "FlatWorld" environment from Figure 1 (left). LTL can easily define some simple objectives, such as safety $G(\neg b)$ (avoid the blue region), reachability $F(g)$ (reach the green region), or progress $F(y)\&X(F(r))$ (reach the yellow, then the red region). We can also combine operators to bring together these objectives into more complex specifications, such as $G(F(r)\&F(y)\&F(g))\&G(\neg b)$, which instructs an agent to oscillate amongst the red, yellow, and green regions indefinitely while avoiding the blue region.

In order to determine the logical satisfaction of an LTL specification, we can transform it into a specialized automaton called a *Limit Deterministic Büchi Automaton (LDBA)*. See Sickert et al. (2016); Hahn et al. (2013); Křetínský et al. (2018) for details on how LTL specifications can be transformed into semantically equivalent LDBA.

More precisely, a (de-generalized) LDBA is a tuple $\mathbb{B} = (\mathcal{B}, \Sigma, T^{\mathbb{B}}, \mathcal{B}^*, \mathcal{E}, b_{-1})$ with a set of states $\mathcal{B}$, the alphabet $\Sigma$ of predicates $\nu$ that defines deterministic transitions in the automaton, a transition function $T^{\mathbb{B}} : \mathcal{B} \times (2^\Sigma \cup \mathcal{E}) \to \mathcal{B}$, a set of accepting states $\mathcal{B}^*$, and an initial state $b_{-1}$. An LDBA has separate deterministic and nondeterministic components $\mathcal{B} = \mathcal{B}_D \cup \mathcal{B}_N$, such that $\mathcal{B}^* \subseteq \mathcal{B}_D$, and for $b \in \mathcal{B}_D$, $x \in \Sigma$ then $T^{\mathbb{B}}(b, x) \subseteq \mathcal{B}_D$. $\mathcal{E}$ is a set of "jump" actions, also known as epsilon-transitions, for $b \in \mathcal{B}_N$ that transitions to $\mathcal{B}_D$ without evaluating any atomic propositions. A path $\xi = (b_0, b_1, \dots)$ is a sequence of states in $\mathcal{B}$ reached through successive transitions under $T^{\mathbb{B}}$, not including the initial state $b_{-1}$, as the labeling function $L^{\mathcal{M}}(s_0)$ transitions the state of $\mathbb{B}$ to $b_0$ to be consistent with the initial state of the MDP.

**Definition 2.1** (Acceptance of $\xi$). We accept a path $\xi = (b_0, b_1, \dots)$ if an accepting state of the Büchi automaton is visited infinitely often by $\xi$.

## 2.2 Problem Statement

We would like to learn a policy that produces satisfactory (accepting) trajectories with respect to a given LTL formula $\varphi$ while maximizing $r$, the reward function from the MDP. Before we define our formal problem statement, we introduce more notation:

**Definition 2.2** (Product MDP). A **product MDP** synchronizes the MDP with an LDBA. Specifically, let $\mathcal{M}^\varphi$ be an MDP with state space $\mathcal{S}^\varphi = \mathcal{S} \times \mathcal{B}$. Policies over our product MDP space can be defined as $\pi : \mathcal{S}^\varphi \to \Delta(\mathcal{A}^\varphi)$, where our new set of actions combine $\mathcal{A}^\varphi((s, b)) = \mathcal{A}(s) \cup \mathcal{E}$, to include the jump transitions in $\mathbb{B}$ as possible actions. We define the space of all possible policies as $\Pi$. The new probabilistic transition relation of our product MDP is defined as:

$$T(s, b, a, s', b') = \begin{cases} T^\mathcal{M}(s, a, s') & a \in \mathcal{A}(s), b' = T^\mathbb{B}(b, L(s')) \\ 1 & a \in \mathcal{E}, b' = T^\mathbb{B}(b, a), s = s' \\ 0 & \text{otherwise} \end{cases} \tag{1}$$

A policy generates trajectories $\tau = ((s_0, b_0, a_0), (s_1, b_1, a_1), \dots)$ in the product MDP. Define $\mathcal{R}(\tau) \equiv \sum_{t=0}^\infty \gamma^t r(s_t, a_t)$ as the total reward along a trajectory $\tau$.

**Definition 2.3** (Trajectory acceptance). A trajectory is said to be **accepting** with respect to $\varphi$ ($\tau \models \varphi$, or "$\varphi$ accepts $\tau$") if there exists some $b \in \mathcal{B}^*$ that is visited infinitely often.

**Definition 2.4** (Policy satisfaction). A policy $\pi \in \Pi$ *satisfies* $\varphi$ with some probability $\mathbb{P}[\pi \models \varphi] = \mathbb{E}_{\tau \sim \mathcal{M}_\pi^\varphi}[\mathbf{1}_{\tau \models \varphi}]$. Here, $\mathbf{1}$ is an indicator variable that checks whether or not a trajectory $\tau$ is accepted by $\varphi$, and $\mathcal{M}_\pi^\varphi$ is the distribution of trajectories induced by policy $\pi$ in a product MDP $\mathcal{M}^\varphi$.

**Definition 2.5** (Probability-optimal policies). We will denote $\Pi^*$ as the set of policies that maximize the probability of satisfaction with respect to $\varphi$; that is, the policies that have the highest probability of producing an accepted trajectory: $\Pi^* = \{\pi \in \Pi | \mathbb{P}[\pi \models \varphi] = \max_{\pi' \in \Pi} \mathbb{P}[\pi' \models \varphi]\}$.

Our aim is to find a policy in the probability-optimal set $\Pi^*$ that collects the largest expected cumulative discounted reward. We state this constrained objective formally as follows:

$$\pi^* \in \text{argmax}_{\pi \in \Pi^*} \mathbb{E}_{\tau \sim \mathcal{M}_\pi^\varphi}[\mathcal{R}(\tau)] \tag{2}$$

For notational convenience, we will refer to the MDP value function as $\mathcal{R}_\pi \equiv \mathbb{E}_{\tau \sim \mathcal{M}_\pi^\varphi}[\mathcal{R}(\tau)]$.

In certain cases, the probability-optimal set of policies $\Pi^*$ may be empty; consequently, a solution to 2 may not exist. In section 3, we introduce a proxy objective with a similar potential of non-existence and discuss how our ultimate optimization objective behaves in this setting.

To align with our intended applications towards deep RL, we consider stochastic, memoryless policies over the product MDP. Such policies are capable of capturing probability-optimal policies for a given LTL specification if an optimal policy exists (Bozkurt et al., 2020; Voloshin et al., 2022).

## 3 LTL-Constrained Policy Optimization

Finding a policy within $\Pi^*$ is, in general, not tractable: an LTL constraint $\varphi$ is defined over infinite-length trajectories but policy rollouts in practice produce only finite-length trajectories (Yang et al., 2022). We adopt *eventual discounting* (Voloshin et al., 2023), a common approach in the existing literature which aims to optimize a proxy value function that approximates the satisfaction of $\varphi$. Eventual discounting is defined as:

$$V_\pi = \mathbb{E}_{\tau \sim \mathcal{M}_\pi^\varphi}\left[\sum_{t=0}^\infty \Gamma_t r_{\text{LTL}}(b_t)\right], \qquad \Gamma_t = \gamma_\varphi^{j_t}, \quad r_{\text{LTL}}(b_t) = \begin{cases} 1 & \text{if } (b_t \in \mathcal{B}^*) \\ 0 & \text{otherwise} \end{cases} \tag{3}$$

where $j_t = \sum_{k=0}^t r_{\text{LTL}}(b_k)$ counts how many times the set $\mathbb{B}^*$ has been visited (up to and including the current timestep). Notably, eventual discounting does not discount based on the amount of time between

visits to an accepting state. A policy that maximizes this eventual discounting reward is approximately probability-optimal with respect to $\varphi$ when $\gamma_\varphi$, the discounting factor associated with $\varphi$, is selected properly (see Theorem 4.2 in Voloshin et al. (2023) for an exact bound).

As a result of eventual discounting, we can replace $\Pi^*$ in objective 2 with the set of policies that maximize $V_\pi$. Let $V_{\max} = \max_{\pi \in \Pi} V_\pi$ be the maximal value.

$$\pi^* = \operatorname*{argmax}_{\pi \in \{\pi \in \Pi \mid V_\pi = V_{\max}\}} \left[ \mathcal{R}_\pi \right] \tag{4}$$

We now form the Lagrangian dual of objective 4 as

$\pi^* = \min_\lambda \operatorname{argmax}_{\pi \in \Pi} \left[ \mathcal{R}_\pi + \lambda(V_\pi - V_{\max}) \right]$. In theorem 3.2 we show that because we only care about constraint-maximizing policies, there exists $\lambda^* \in \mathbb{R}$ such that solving the inner maximization of the Lagrangian dual must be constraint optimal for any fixed $\lambda > \lambda^*$. Intuitively, the higher $\lambda$ is, the more our learned policy will account for $V_\pi$ during optimization until the constraint must be satisfied. At that point, because we are already achieving the maximum possible $V_\pi$, any additional lift will only come from maximizing over the MDP value $\mathcal{R}$, even if we continue to increase $\lambda$. With this observation, we can form an unconstrained objective function from objective 4 to be the following:

$$\pi^* = \arg\max_{\pi \in \Pi} \left[ \mathcal{R}_\pi + \lambda V_\pi \right] \tag{5}$$

where we have dropped the dependence on $V_{\max}$ since it is a constant and fixed $\lambda > \lambda^*$. We show that under certain assumptions, an exact value for $\lambda^*$ can be found to ensure that a policy that maximizes eq. 5 will certainly maximize $V_\pi$.

**Assumption 3.1.** *There exists a positive nonzero gap $\epsilon > 0$ between the value $V_\pi$ of policies in $\pi \in \Pi^*$ and the highest-value policies that are not; that is, $V_{\max} - \max_{\pi \in (\Pi \setminus \Pi^*)}(V_\pi) > \epsilon$.*

**Theorem 3.2.** *Under Assumption 3.1, for any choice of $\lambda > \frac{R_{\max} - R_{\min}}{\epsilon(1-\gamma)}$, the solution to objective 5 must be a solution to objective 4. See Appendix Section B for the proof.*

We note that Assumption 3.1 can be found in previous literature (Voloshin et al., 2023) and serves as a sufficient but not necessary condition for our results. We provide further analysis for the existence of Assumption 3.1 in Appendix Section B.1.

As briefly mentioned in Section 2, the probability-optimal set of policies with respect to $\varphi$ may be empty. The same is true for our updated definition of $\Pi^*$ that contain policies that achieve $V_{\max}$. In the case of this non-existence, Assumption 3.1 does not hold, and a policy that optimizes 5 will prioritize improving $V_\pi$ at the potential expense of $\mathcal{R}_\pi$. We provide an extended discussion of this consequence in Section B.2.

**Empirical Considerations.** Since the conditions for Assumption 3.1 are often unknown, there may not be a verifiable way to know that that our learned policy is maximizing $V_\pi$. Because of this, we will treat $\lambda$ as a tunable hyperparameter that allows a user to trade off the relative importance of empirically satisfying the LTL constraint. There are a number of strategies one can use to find an appropriate $\lambda$: for example, one can iteratively increase $\lambda$ until a desired LTL reward is achieved. In our experiments, we show an example of this trade off, and notice that the trade off lessens in severity once $\lambda$ exceeds a value that enables learning LTL-satisfying policies (table 2).

## 4 Cycle Experience Replay (CyclER)

To distinguish between the MDP's reward function and the eventual-discounting proxy reward in 3, we write the MDP reward function $r(s, a)$ as $r_{\mathrm{MDP}}(s, a)$. In Deep RL settings, we maximize objective 5 using the reward function $r_{\mathrm{DUAL}}(s_t, b_t, a_t) = \gamma^t r_{\mathrm{MDP}}(s_t, a_t) + \Gamma_t \lambda r_{\mathrm{LTL}}(b_t)$.

However, optimizing objective 5 is challenging due to the sparsity of $r_{\mathrm{LTL}}$. $r_{\mathrm{LTL}}$ is nonzero only when an accepting state in $\mathbb{B}$ is visited, which may require a long, precise sequence of actions.

Consider the FlatWorld MDP and LDBA in Figure 1. The MDP's reward function incentivizes visiting the small purple regions in the world. Under $r_{\text{LTL}}$, a policy will receive no reward until it completes the entire task of avoiding blue and visiting the red, yellow, and green regions through random exploration. If $r_{\text{MDP}}$ is dense, a policy may fall into an unsatisfactory 'local optimum' by optimizing for $r_{\text{MDP}}$ it receives early during learning, and ignore $r_{\text{LTL}}$ entirely. In Figure 2, we see that a policy trained on $r_{\text{DUAL}}$ makes such an error.

We seek to address this shortcoming by *automatically* shaping $r_{\text{LTL}}$ so that a more dense reward for $\varphi$ is available during training. Below, we present our approach, which exploits the known structure of the LDBA $\mathbb{B}$ and cycles within $\mathbb{B}$ that visit accepting states.

### 4.1 Rewarding Accepting Cycles in $\mathbb{B}$

By definition of LTL satisfaction (def. 2.3), a trajectory must repeatedly visit an accepting state $b^*$ in an LDBA. In the context of the automaton itself, that means that an accepting trajectory will traverse an *accepting path* from the initial state to an accepting state, and then repeatedly traverse *accepting cycles* within $\mathbb{B}$ that continually visit accepting states.

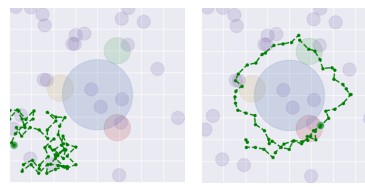

Figure 2: Trajectories from unshaped $r_{\text{DUAL}}$ (left) and CyclER $r_{\text{DUAL}}$ (right) for the formula $G(F(r)\&F(g)\&F(y)) \ \& \ G(\neg b)$.

**Definition 4.1** (Accepting Initial Path (AIP)). An accepting initial path in $\mathbb{B}$ is a set of valid transitions $(b_i, \nu, b_j)$ (i.e., the predicate $\nu$ that transitions $b_i$ to $b_j$) in $\mathbb{B}$ that starts at the initial state $b_{-1}$ and ends at an accepting state $b_k^* \in \mathcal{B}^*$.

**Definition 4.2** (Accepting Cycle (AC)). An accepting cycle in $\mathbb{B}$ is a set of valid transitions $(b_i, \nu, b_j)$ in $\mathbb{B}$ that start and end at accepting states $b_k^*, b_l^* \in \mathcal{B}^*$.[1]

Our key insight is that we can use accepting paths and cycles in $\mathbb{B}$ to shape $r_{\text{LTL}}$. Instead of only providing reward when an accepting state in $\mathbb{B}$ is visited (as per previous approaches e.g. Voloshin et al. (2023)), we reward progress within an accepting path or cycle. In our example from fig 1, if we reward each transition in the initial path $\{1, 2, 3, 0\}$ and the cycle with the same states, the agent would receive rewards for visiting the red region, then yellow, then green, then for returning to red, and so on.

Multiple accepting paths and cycles may exist in $\mathbb{B}$. The path and cycle that is used to shape $r_{\text{LTL}}$ cannot be picked arbitrarily, since they may be infeasible under the dynamics of the MDP. For example, the cycle $\{1, 2, 0\}$ in Figure 1 cannot effectively shape $r_{\text{LTL}}$ because it is impossible to be both in the yellow and green regions at the same time.

### 4.2 Reward Shaping with CyclER

CyclER is a reward function that automatically selects paths and cycles to shape $r_{\text{LTL}}$ based on collected experience.

**Definition 4.3** (Minimal AIP (MAIP)). A minimal accepting initial path for accepting state $b_k^*$ is an AIP that does not contain a subcycle for any node $b_i$ in the path where $b_i \neq b_k^*$.

**Definition 4.4** (Minimal AC (MAC)). A minimal accepting cycle $c$ for accepting states $b_k^*$ and $b_l^*$ is an AC that does not contain a subcycle for any node $b_i$ in the cycle where $b_i \notin \{b_k^*, b_l^*\}$.

We provide CyclER with all MAIPs and MACs in an LDBA using Depth-First Search with backtracking (see Appendix Algs. 2 and 3). Let $\mathcal{P}$ and $\mathcal{C}$ be the set of MAIPs and MACs, respectively.

We also maintain a frontier $e$ of visited transitions in $\mathbb{B}$ at each timestep in a trajectory that ensures reward will only be given once per transition until an accepting state is visited. In particular, we set $e[(b_i, \nu, b_i)] = 1$ when a transition $(b_i, \nu, b_i)$ is taken and reset all $e \equiv 0$ when $b_j \in \mathcal{B}^*$. Policies that use CyclER observe $s$, $b$, and $e$ as their current state.

Now we describe the CyclER reward computation. We first collect a full trajectory $\tau$ induced by $\mathcal{M}_\pi^\varphi$ for a given policy $\pi$. Then, at each timestep $t$ from 0 to $|\tau| - 1$, we compute $r_\mathcal{C}$ for every path in $\mathcal{P}$ if we have not

---

[1]Our usage of the word "cycle" is not a cycle in the traditional sense of graph search, but instead refers to paths that connect two accepting states in $\mathbb{B}$ (allowing for "cyclical" acceptance).

yet visited an accepting state, or every cycle in $\mathcal{C}$ if we have. We will abuse notation slightly and use $c$ to refer to elements in either $\mathcal{P}$ or $\mathcal{C}$:

$$r_{\mathcal{C}}(s, b, s', b', e, c) = \begin{cases} \frac{1}{|c|} & \text{if } (b, L^{\mathcal{M}}(s'), b') \in c \text{ and } e[b, L^{\mathcal{M}}(s'), b'] = 0 \\ 0 & \text{otherwise} \end{cases} \tag{6}$$

This function rewards every transition taken in a given $c$. In other words, when an agent "gets closer" to an accepting state by progressing along a path or cycle, we reward that progress. Importantly, we do not reward transitions taken in a given $c$ more than once per visit to an accepting state. In other words, a trajectory that repeatedly takes transitions in $c$ but never reaches an accepting state $b^*$ will only receive reward for the first instance of each transition taken in $c$. To account for $c$ of varying length, rewards are normalized by the length $|c|$.

---

**Algorithm 1:** Cycle Experience Replay (**CyclER**)

**Input:** Trajectory $\tau$, $\mathcal{B}^*$, cycles $\mathcal{C}$, paths $\mathcal{P}$, Labelling function $L^{\mathcal{M}}$

Initialize matrix $R_{\mathcal{C}}$ size $\max(|\mathcal{C}|, |\mathcal{P}|) \times (|\tau| - 1)$;
Initialize $r_{\text{CyclER}}$ to an array of size $(|\tau| - 1)$;
Initialize $j = 0$;
**foreach** $t = 0, \ldots, |\tau| - 1$ **do**
  **if** $j = 0$ **then**
    **foreach** *Path* $p_i \in \mathcal{P}$ **do**
      $R_{\mathcal{C}}[i, t] = r_{\mathcal{C}}(b_t, s_{t+1}, b_{t+1}, e_t, p_i)$
  **else**
    **foreach** *Cycle* $c_i \in \mathcal{C}$ **do**
      $R_{\mathcal{C}}[i, t] = r_{\mathcal{C}}(b_t, s_{t+1}, b_{t+1}, e_t, c_i)$
  Set $e[b_t, L^{\mathcal{M}}(s_{t+1}), b_{t+1}] = 1$;
  **if** $b_{t+1} \in \mathcal{B}^*$ *or* $t + 1 = |\tau|$ **then**
    Select $i = \arg\max_{i \in |C|}(\sum_{j'=j}^{t} R_{\mathcal{C}}[i, j'])$;
    **foreach** $t'$ *from* $j$ *to* $t + 1$ **do**
      $r_{\text{CyclER}}[t'] = R_{\mathcal{C}}[i, t']$
    $j = t + 1$;
    Set all $e \equiv 0$;
**return** $r_{CyclER}$;

---

If we visit an accepting state $b^*$ or reach the end of a trajectory, we retroactively assign rewards to the timesteps that preceded this point, up to the most recent accepting state visit (if one exists). Assigned rewards correspond to the cycle with the highest total reward for that partial trajectory. Put simply, CyclER picks the 'best' cycle for a partial trajectory and uses it to shape reward. Even if a trajectory does not manage to visit an accepting state, CyclER will still provide reward if it was able to take *any* transition along *any* MAIP. The CyclER algorithm is formally outlined in Algorithm 1.

Let us walk through an example of using CyclER to shape reward by referring to the specification $\varphi$ and trajectory in the FlatWorld MDP from Figure 1. Here, the agent produces a trajectory of length 4, indicated by the end of each line segment, starting from its initial state. The corresponding LDBA path for this trajectory $\xi$ is $(1, 2, 2, 3)$. Under the unshaped (non-CyclER) proxy reward defined in 3, no reward would be gained from this trajectory, as the accepting state of the LDBA is not visited. Instead, let us consider how CyclER shapes the reward for each transition in this trajectory. There are three MAIPs in this LDBA: $\{(1, ryg, 0)\}$, $\{(1, r, 2), (2, yg, 0)\}$, and $\{(1, r, 2), (2, g, 3), (3, y, 0)\}$. For each of these MAIPs, CyclER will retroactively assign rewards to each transition within the trajectory. Under the MAIP $\{(1, ryg, 0)\}$, no transitions will receive reward, as the predicate $ryg$ is not achieved. For $\{(1, r, 2), (2, yg, 0)\}$, the transition to the second timestep of the trajectory will receive a reward of $\frac{1}{2}$ for visiting the red region. Lastly, for the MAIP $\{(1, r, 2), (2, g, 3), (3, y, 0)\}$, both the transitions to the second and fourth timesteps will receive a reward of $\frac{1}{3}$, for visiting the red and green regions, respectively. Therefore, CyclER will choose to shape the trajectory's rewards using the MAIP $\{(1, r, 2), (2, g, 3), (3, y, 0)\}$, as it results in the highest cumulative reward assigned to the entire trajectory.

We denote the rewards returned from Algorithm 1 as $r_{\text{CyclER}}$. $r_{\text{CyclER}}$ can be used in place of the unshaped $r_{\text{LTL}}$ in function 3 to provide a more dense reward for $\tau$.

**Theorem 4.1** (Informal). *By replacing $r_{LTL}$ with $r_{CyclER}$ in 3, the solution to problem 5 remains (approximately) probability optimal in satisfying the LTL formula $\varphi$. See Appendix Lemma C.3 for the proof.*

### 4.3 CyclER with Quantitative Semantics

A number of recent works have explored the usage of *Quantitative Semantics* (QS) to help shape rewards for temporal logic tasks (Li et al., 2017; Balakrishnan & Deshmukh, 2019; Jothimurugan et al., 2021; Kalagarla et al., 2021; Ikemoto & Ushio, 2022). QS defines a set of rules for temporal logic which extend Boolean logic to

operations over real values. Using QS, we can take real-valued signals from our atomic propositions $x \in \text{AP}$, and compose them with the QS version of our logical connectives ($\&$, $\neg$ and $\rightarrow$) and temporal operators ($(X)$, $(G)$, $(F)$, and $(U)$) to compute a real-valued signal for how close a trajectory comes to satisfying a specification. We will refer to this computation as the *quantitative evaluation* of a trajectory with respect to an LTL task. The exact quantitative semantics of the aforementioned LTL operations are provided in Figure 9 of the Appendix.

Unfortunately, there are several shortcomings of "off-the-shelf" usage of QS as a reward function. Quantitative evaluation produces a single value for an entire trajectory, which makes credit assignment for individual transitions difficult. More pressingly, quantitative evaluation of a finite trace frequently produces values that do not correlate with visits to accepting states in $\mathbb{B}$, especially for LTL formulae with indefinite horizons or arbitrarily ordered sub-goals. Existing approaches have circumvented these issues by using QS for explicitly time-bounded temporal logics with simple tasks (Kalagarla et al., 2021; Balakrishnan & Deshmukh, 2019), or by considering fragments of LTL that can be reasoned about as finite sequences of ordered sub-tasks (Jothimurugan et al., 2021). In what follows, we show that CyclER easily incorporates QS for more effective LTL reward shaping by considering each transition in $\mathbb{B}$ as an independently evaluable sub-task.

We first define some notation. In order to use QS, we assign *robustness measures* (real-valued signals) $f_x : \mathcal{S} \rightarrow \mathbb{R}$ to each atomic proposition $x \in \text{AP}$, where $x$ is true when $f_x(s) \geq c_x$ (a constant threshold). We will follow standard practice and notate the quantitative evaluation of an LTL formula $\varphi$ over a trajectory $\tau$ as $\rho_\varphi(\tau)$, where $\varphi$ is true when $\rho_\varphi(\tau) > 0$. We also define a maximum and minimum for $\rho$ as $\rho_{\max}$ and $\rho_{\min}$, respectively.

Now we explain how to incorporate QS into CyclER. At a given state $b$ in $\mathbb{B}$, we can think of our sub-task as taking the next transition in the accepting path or cycle currently under consideration by CyclER. We need to only consider one transition at a time because CyclER reasons about each accepting path and cycle independently. Our approach to incorporating QS builds on this idea by rewarding quantitative *progress* towards taking the next transition in a path or cycle. Importantly, each transition $\nu$ in $\mathbb{B}$ is associated with an atomic predicate, which can be quantitatively evaluated at individual states rather than entire trajectories (i.e., $\rho_\nu(s) : \mathcal{S} \rightarrow \mathbb{R}$). If we move from state $s$ to $s'$ in $\mathcal{M}$, we can evaluate how much closer we are to satisfying the predicate of our next transition $\nu$ by taking the difference in quantitative evaluation between successive states: $\rho_\nu(s') - \rho_\nu(s)$. This measure of progress is used to shape reward.

Specifically, our approach to incorporating QS uses the following reward function for a given cycle or initial path $c$. We use $c[b]$ to refer to the transition predicate in $c$ with parent node $b$:

$$r_{\text{qs}}(s, b, s', b', e, c) = \begin{cases} \frac{\rho_{c[b]}(s') - \rho_{c[b]}(s)}{(\rho_{\max} - \rho_{\min}) * |c|} & \text{if } (b, L^{\mathcal{M}}(s'), b') \in c \text{ and } e[b, L^{\mathcal{M}}(s'), b'] = 0 \\ 0 & \text{otherwise} \end{cases} \tag{7}$$

We use $\rho_{\max}$ and $\rho_{\min}$ to normalize the quantitative progress made towards taking a transition[2]. Reward function 7 acts as a direct substitute to function 6 in Alg. 1 to compute $r_{\text{CyclER}}$. Unlike function 6, function 7 allows for nonzero rewards to be given more than once per transition in an LDBA, allowing for dense reward even in LDBAs with few transitions.

In our experiments (section 5), we show that incorporating quantitative semantics into CyclER for shaping $r_{\text{LTL}}$ leads to improvement in empirical performance when compared to existing methods of using QS. We provide the full QS for LTL, along with additional explanation and examples for CyclER+QS in Appendix E.

## 5 Experiments

We demonstrate experimental results in several domains with continuous state and action spaces on LTL tasks of varying complexity. We seek to answer the following questions: **(1)** Does CyclER learn satisfactory policies

---

[2]We note that the reward in fn. 7 is most well-shaped when the robustness measures for all $x \in \text{AP}$ are of similar scale, but we make no such assumptions for the sake of generality.

and avoid ignoring $r_{\text{LTL}}$ in favor of $r_{\text{MDP}}$? **(2)** Does a policy that optimizes the dual reward formulation in $r_{\text{DUAL}}$ gain higher $r_{\text{MDP}}$ than a policy that only seeks to satisfy the LTL constraint? **(3)** How does the value of $\lambda$ in $r_{\text{DUAL}}$ affect the performance of the learned policy?

### 5.1 Experimental Domains and Tasks

In our experiments, we evaluate the efficacy of CyclER on indefinite-horizon ($\omega$-regular) tasks expressible by LTL. We use environments where $r_{\text{MDP}}$ does not explicitly correlate with $r_{\text{LTL}}$ in order to effectively distinguish between policies that learn to only optimize $r_{\text{MDP}}$ and policies that learn to satisfy the LTL specification.

**FlatWorld** The FlatWorld domain (1) is a two dimensional world with continuous state and action spaces. The agent (denoted by a green dot) starts at (-1, -1). The agent's state, denoted by x, is updated by an action $a$ via $x' = x + a/10$ where $x \in \mathbb{R}^2$ and $a \in [0, 1]^2$. There exists a set of randomly generated purple 'bonus regions', which offer a small reward when visited. We use the specification from Figure 1 as our LTL task.

**ZonesEnv** We use the Zones environment from the MuJoCo-based Safety-Gymnasium suite of environments (Ji et al., 2023). In this domain, a robot must navigate the environment, which includes four differently colored goal regions and 'hazard' areas that offer a small negative reward. The robot receives an observation of lidar data that detects the presence of nearby objects at each timestep. The LTL task description instructs the agent to oscillate amongst visiting the four colored regions.

**ButtonsEnv** We use the Buttons environment, also from Safety-Gymnasium. This domain is a more challenging version of the Zones environment, where an agent must press a number of small buttons in a larger space while avoiding cube-shaped 'gremlins' that move in a fixed circular path. The LTL task description instructs the agent to press two specific buttons infinitely often, while avoiding making contact with gremlins. Unlike the ZonesEnv, 'bonus' regions are scattered around the environment, offering a small reward if visited.

### 5.2 Implementation Details and Baselines

We use entropy-regularized PPO (Schulman et al., 2017) with a Gaussian policy over the action space as our policy class.

Although we are not aware of an existing approach that considers reward optimization under general LTL constraints for deep RL, we compare against a number of existing reward methods for temporal logic-guided RL as $r_{\text{LTL}}$ in our $r_{\text{DUAL}}$ formulation. We use a baseline policy trained using the LCER method (Voloshin et al., 2023), a state-of-the-art approach to RL for general LTL that uses an unshaped reward with counterfactual experience replay to improve the sample efficiency of learning. Additionally, we compare against the LCER baseline, but trained *only* on the unshaped LTL reward function $r_{\text{LTL}}$, in order to observe the performance of a policy that does not get 'distracted' during training by $r_{\text{MDP}}$.

In the ZonesEnv and ButtonsEnv domains, where the dynamics are more complex, we define simple robustness measures for each atomic proposition and **use the QS version of CyclER defined in section 4.3**. In these environments, we compare against two additional baselines that also use QS for reward shaping and are computable for infinite-horizon LTL tasks: a TLTL-based reward (Li et al., 2017) and BHNR (Balakrishnan & Deshmukh, 2019).

For each baseline, $\lambda$ was chosen to be that which led to best performance (on unshaped $r_{\text{LTL}}$, using $r_{\text{MDP}}$ as a tie-breaker) from a hyperparameter sweep. The robustness measures used in these domains along with all hyperparameters used during training are available in Appendix G.

### 5.3 Results

**(1) Does CyclER learn satisfying policies and prevent ignoring $r_{\text{LTL}}$?** Yes - our results demonstrate that CyclER achieves significant improvement in performance in satisfying the LTL task when compared to our baseline methods. In Figure 3, we plot the learning curves for both the *unshaped* $r_{\text{LTL}}$ and $r_{\text{MDP}}$. We

| | FlatWorld | | ZonesEnv | | ButtonsEnv | |
|---|---|---|---|---|---|---|
| | $\mathcal{B}^*$ visits | $r_{\mathrm{MDP}}$ | $\mathcal{B}^*$ visits | $r_{\mathrm{MDP}}$ | $\mathcal{B}^*$ visits | $r_{\mathrm{MDP}}$ |
| CyclER | $\mathbf{2.0 \pm 0.5}$ | $45.3 \pm 8.5$ | $\mathbf{1.8 \pm 0.4}$ | $-27.8 \pm 4.55$ | $\mathbf{2.6 \pm 0.3}$ | $30.4 \pm 5.6$ |
| LCER | $0.0 \pm 0.0$ | $103.4 \pm 76.6$ | $0.0 \pm 0.0$ | $-3.8 \pm 1.9$ | $0.0 \pm 0.0$ | $118.8 \pm 143.2$ |
| LCER, no $r_{\mathrm{MDP}}$ | $0.8 \pm 0.4$ | $30.8 \pm 10.1$ | $0.0 \pm 0.0$ | $-2.7 \pm 0.9$ | $0.6 \pm 0.4$ | $13.2 \pm 8.53$ |
| TLTL | - | - | $0.0 \pm 0.0$ | $-4.0 \pm 2.2$ | $0.0 \pm 0.0$ | $9.6 \pm 6.7$ |
| BHNR | - | - | $0.0 \pm 0.0$ | $-0.8 \pm 0.6$ | $0.0 \pm 0.0$ | $35.8 \pm 7.9$ |

Table 1: Reward average and standard deviation achieved on each domain with an extended horizon. $\mathcal{B}^*$ visits identifies the average number of visits to an accepting state in $\mathbb{B}$ achieved for a trajectory from $\pi$, and $r_{\mathrm{MDP}}$ refers to the average MDP reward collected during a trajectory.

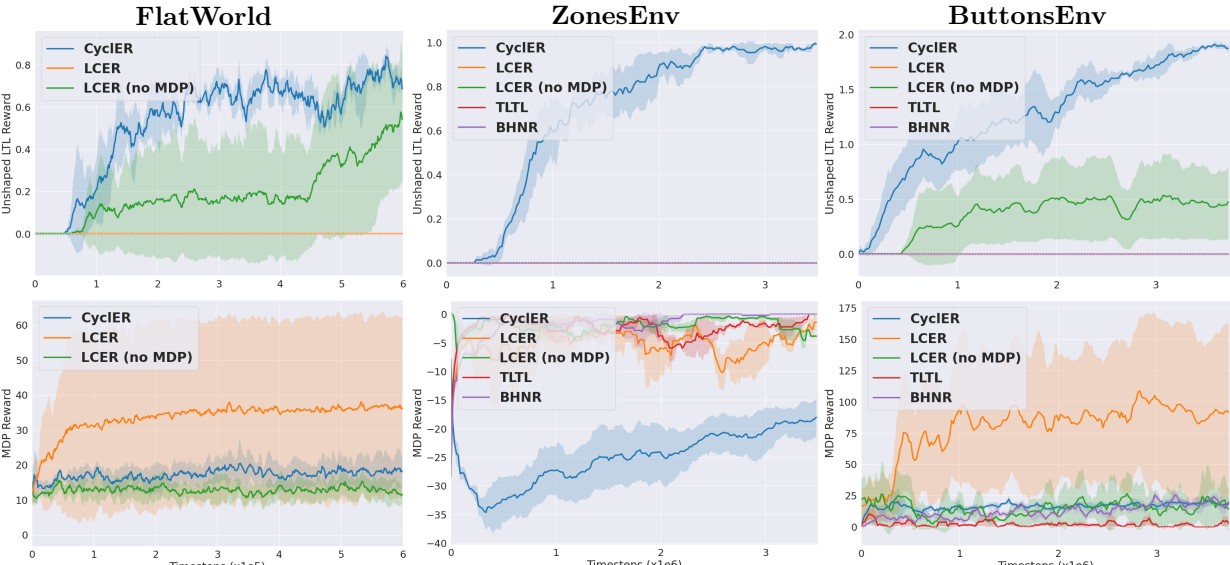

Figure 3: Training curves showing unshaped $r_{\mathrm{LTL}}$ (top) and $r_{\mathrm{MDP}}$ (bottom) performance averaged over 5 random seeds. Each point is the mean of 10 stochastic policy rollouts.

record the (stochastic) performance of the best policies found during training on an extended horizon to enable repeated visits to the accepting state, and present the results (averaged over 50 rollouts) in Table 1.

We find that unshaped rewards (LCER) quickly ignore $r_{\mathrm{LTL}}$ in most trials. From Table 1, we see that the LCER baseline even without $r_{\mathrm{MDP}}$ was not able to accomplish the LTL task as consistently as CyclER, even when successful in visiting an accepting state. This implies that reward shaping is critical for LTL-guided RL even in settings where no $r_{\mathrm{MDP}}$ is present. In ZonesEnv and ButtonsEnv, the TLTL and BHNR baselines, which are not suited to infinite-horizon tasks with multiple unordered subgoals, learned behavior that optimized their respective QS-shaped LTL rewards but did not correlate with task satisfaction (zero $r_{\mathrm{LTL}}$ achieved in Figure 3). CyclER with QS, on the other hand, quickly learned to achieve the tasks in these two domains. Most meaningfully, CyclER is able to repeatedly visit the accepting state in all domains (as evidenced by Table 1), demonstrating that our technique enables consistent-behaving policies that can indefinitely traverse accepting cycles. We additionally provide a qualitative analysis of learned behavior in the FlatWorld domain in Appendix D.

**(2) Does optimizing $r_{\mathrm{DUAL}}$ improve $r_{\mathrm{MDP}}$?** To evaluate this question, we conducted an ablation study where we trained a CyclER-based policy in the FlatWorld domain, making it completely unaware of $r_{\mathrm{MDP}}$, and then evaluated its performance to observe if the $r_{\mathrm{DUAL}}$ formulation led to a nontrivial difference in behavior between policies. We found that the $r_{\mathrm{DUAL}}$-optimizing CyclER policy visited $\mathcal{B}^*$ an average of 2.0 times (std. dev. 0.3) and achieved an MDP reward of 45.3 (std. dev. 8.5), and the $r_{\mathrm{LTL}}$-only policy visited $\mathcal{B}^*$ an average of of 2.2 times (std. dev. 0.4) and achieved an MDP reward of 27.4 (std. dev. 0.7). Optimizing

$r_{\mathrm{DUAL}}$ does lead to an improvement in $r_{\mathrm{MDP}}$, albeit at the potential cost of LTL satisfaction. We provide a qualitative comparison of the two learned policies in Appendix D.

**(3) How does varying $\lambda$ affect the resulting policy?** In Table 2, we report results from a study where we vary the value of $\lambda$ in $r_{\mathrm{DUAL}}$ for the FlatWorld domain experiment. We observe, as expected, a tradeoff in the performance of LTL satisfaction and $r_{\mathrm{MDP}}$ as $\lambda$ increases. However, we notice that the tradeoff diminishes once a value for $\lambda$ is reached that enables LTL-satisfying behavior. This supports our intuition that $\lambda$ can effectively be used as a hyperparameter to trade off the empirical performance of LTL satisfaction and the MDP reward achieved by a policy.

| | **FlatWorld** | |
|---|---|---|
| | $\mathcal{B}^*$ visits | $r_{\mathrm{MDP}}$ |
| $\lambda = 100$ | $0.0 \pm 0.0$ | $62.7 \pm 4.36$ |
| $\lambda = 200$ | $0.0 \pm 2.0$ | $69.6 \pm 4.1$ |
| $\lambda = 300$ | $2.0 \pm 0.4$ | $37.2 \pm 9.0$ |
| $\lambda = 400$ | $2.1 \pm 0.4$ | $34.3 \pm 7.3$ |

Table 2: Performance results for CyclER with differing $\lambda$.

## 6 Related Work

Our work falls under the broad umbrella of *specification-guided* approaches to *correct-by-construction* machine learning (Seshia et al., 2022), with a particular focus on RL.

**Temporal Logic-Constrained Policy Optimization.** Previous work has explored cost-optimal control under linear temporal logic constraints with known dynamics (Ding et al., 2014; Cai et al., 2021). More recently, interest has emerged in RL-based approaches to logic-constrained policy optimization. Voloshin et al. (2022) provides an exact solution method for policy optimization under general LTL constraints in discrete settings where the dynamics are unknown by assuming a lower bound on transition probabilities in $\mathcal{M}$. Other works focus on Signal Temporal Logic (STL) and are either designed for discrete spaces (Kalagarla et al., 2021) or lack guarantees (Ikemoto & Ushio, 2022). Our CyclER-QS method can be viewed as an approach designed for a subset of STL, with improved performance by our leveraging of the LDBA during policy learning.

**RL with Temporal Logic Objectives.** In contrast to settings with both temporal logic constraints and reward functions, a significant amount of work has been devoted to developing RL approaches with temporal logic specification(s) as the lone objective. Early efforts focused primarily on using Q-learning-style methods over augmentations of $\mathcal{M}$ (Sadigh et al., 2014; Aksaray et al., 2016; Venkataraman et al., 2020; Cai et al., 2021). Subsequent works (Hasanbeig et al., 2020; Toro Icarte et al., 2022; Camacho et al., 2019; Jothimurugan et al., 2019) extend temporal logic-guided RL to deep RL settings. In developing the theoretical limitations of temporal-logic guided RL, (Yang et al., 2022; Alur et al., 2022) show that guarantees on RL for LTL cannot in general be made. To obtain (approximate) guarantees on learning, existing works have made assumptions on the environment dynamics (Fu & Topcu, 2014; Voloshin et al., 2022; Wolff et al., 2012) or finitized the policy's horizon through discounting or recurrence time (Alur et al., 2023; Perez et al., 2023). In continuous spaces, prior works provide guarantees that the optimal policy under a proxy objective will satisfy the original logical specification of interest (Voloshin et al., 2023; Hasanbeig et al., 2020; Jothimurugan et al., 2021; Camacho et al., 2019), similar to the guarantees made in our work.

To handle longer-horizon specifications, previous endeavors proposed compositional RL approaches that leverage the DAG-like structure for finitary fragments of temporal logic (Jothimurugan et al., 2021; Bonassi et al., 2023). Other works take a multi-task RL approach that learns subtasks, which allows for the completion of extended-horizon tasks and unseen tasks over the same AP set (Vaezipoor et al., 2021; Qiu et al., 2023; León et al., 2022; Liu et al., 2022). An extensive body of work uses quantitative signals to provide denser feedback for temporal logic objectives that aids policy learning in longer-horizon task settings Li et al. (2017); Balakrishnan & Deshmukh (2019); Ikemoto & Ushio (2022); Krasowski et al. (2023). These quantative signals, which are often captured in Signal Temporal Logic, are also used by our QS approach introduced in section 4.3. However, the usage of these quantitative signals leads to a number of pitfalls in deep RL that CyclER is able to avoid, as discussed in section 4.3 and Appendix E. More recent work considers problem settings where there is uncertainty in an agent's knowledge of atomic propositions and proposes a belief-based approach to policy learning in this setting (Li et al., 2024). Our approach is able to handle indefinite-horizon specifications for single tasks and we see the integration of our reward shaping into both multi-task frameworks and noisy environments as exciting directions for future work.

**Constrained Policy Optimization.** The broader constrained policy optimization works mostly relate to the constrained Markov Decision Process (CMDP) framework (Le et al., 2019; Achiam et al., 2017; Altman, 2021), which enforce penalties over expected constraint violations rather than absolute constraint violations. In contrast, our work aims to satisfy absolute constraints in the form of LTL.

## 7 Conclusion

This paper proposes a novel approach to finding policies that are both reward-maximal and probability-optimal with respect to an LTL constraint. Specifically, we introduce CyclER, an experience replay technique that automatically shapes the LTL proxy reward based on cycles within a Büchi automaton, alleviating a sparsity issue that often plagues LTL-driven RL approaches. CyclER enables LTL-constrained policy optimization in continuous spaces using function approximators. We extend CyclER to effectively use quantitative semantics for full LTL and demonstrate its success empirically.

There are numerous directions for future work. For example, the reward shaping idea behind CyclER can be extended to other classes of logical specifications, such as Reward Machines (Toro Icarte et al., 2022). We are also interested in applying CyclER to accelerate learning in multi-task LTL settings, such as (Vaezipoor et al., 2021; Qiu et al., 2023).

### Acknowledgments

The authors would like to thank Adwait Godbole, Federico Mora and Niklas Lauffer for their helpful feedback. This work was supported in part by an NDSEG Fellowship (for Shah), ONR Award No. N00014-20-1-2115, DARPA contract FA8750-23-C-0080 (ANSR), C3DTI, Toyota and Nissan under the iCyPhy center, and by NSF grant 1545126 (VeHICaL).

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

## Contents

## A   Limitations

The success of CyclER is ultimately limited to the quality and achievability of the atomic propositional variables in a given environment. If robustness measures are not available and a task specification has relatively few variables that are difficult to satisfy, CyclER will not provide significant improvement over existing unshaped LTL reward proxies. For example, the specification $F(G(x))$ with no robustness measure for $x$ will offer the same reward under CyclER and existing methods. When robustness measures are available, it is important that measures for each variable in the set AP are of a similar scale, so that the QS-shaped rewards do not vary highly across different transitions in $\mathbb{B}$. The issue of needing similar scale for robustness measures is well-known in the temporal logic literature and is an open direction for future work Balakrishnan & Deshmukh (2019); Li et al. (2017).

Although we show that the performance of policy learning is somewhat robust to $\lambda$ in our formulation of $r_{\text{DUAL}}$, we do not have a systematic way of finding an appropriate value for $\lambda$ beyond traditional hyperparameter search methods. We see an interesting opportunity for future work to intelligently search for $\lambda$ based on the desired $r_{\text{LTL}}$ and $r_{\text{MDP}}$ of a user.

The CyclER approach incurs computational overhead by (1) computing all possible cycles prior to policy learning and (2) keeping track of all potential reward values for each cycle for each trajectory stored in an agent's replay buffer. We did not observe a significant slowdown or memory increase in our experiments as a result of this overhead. We acknowledge that in complex specifications with a large number of cycles this overhead may become meaningful.

## B   Proof for Theorem 3.2

*Proof.* Consider two policies: (1) $\pi \in \Pi \setminus \Pi^*$, which does not achieve $V_{\max}$, (2) $\tilde{\pi} \in \Pi^*$, achieving $V_{\max}$. Let $R_{\max}$ and $R_{\min}$ be upper and lower bounds on the maximum and minimum achievable single-step reward in $\mathcal{M}$, respectively. Evaluating objective 5 for both of these policies satisfies the following series of inequalities:

$$\mathcal{R}_{\tilde{\pi}} + \lambda V_{\tilde{\pi}} \overset{(a)}{\geq} \frac{R_{\min}}{1-\gamma} + \lambda(V_\pi + \epsilon) \overset{(b)}{\geq} \frac{R_{\max}}{1-\gamma} + \lambda V_\pi \overset{(c)}{\geq} \mathcal{R}_\pi + \lambda V_\pi$$

where $(a)$ follows from assumption 3.1 and bounding the worst-case MDP value, $(b)$ follows from selecting $\lambda > \frac{R_{\max} - R_{\min}}{\epsilon(1-\gamma)} (\equiv \lambda^*)$, $(c)$ follows since the highest MDP value achievable by $\pi$ must be upper bounded by the best-case MDP value.

As a consequence of $(a-c)$ we see that policies achieving $V_{\max}$ are preferred by objective 5. Consider $\pi^* \in \Pi^*$, the solution to objective 5. Thus, since $\pi^* \in \Pi^*$, then $\pi^*$ must also achieve $V_{\pi^*} = V_{\tilde{\pi}} = V_{\max}$. Therefore, in comparing objective 5 for both $\pi^*$ and $\tilde{\pi}$ it follows immediately that $\mathcal{R}_{\pi^*} \geq \mathcal{R}_{\tilde{\pi}}$ since $\pi^*$ is optimal for objective 5. Since the choice of $\tilde{\pi}$ is arbitrary, we have shown that $\pi^*$ is also a solution to objective 4.   $\square$

### B.1   On the existence of Assumption 3.1.

If we are restricted to stationary policies and the space of policies $\Pi$ is finite, then assumption 3.1 will always hold. A finite space of policies can be enumerated over, and we can take the difference between the optimal and next-best policies to find $\epsilon$. As an example, consider a toy MDP with a continuous, 1-dimensional state space $[0, 1]$ and continuous, 1-dimensional action space $[0, 1]$, where the transition function determines the next state as the agent's action, i.e. $T^{\mathcal{M}}(s, a, s') = a$. Suppose we are given a task specification $G(F(1))$. Under this specification, the agent will receive a reward of 1 every time it outputs 1, and 0 otherwise.

Consider a finite-sized policy class $\Pi$ of just two deterministic policies: $\pi_0$ that only outputs 0, and $\pi_1$ that only outputs 1. Here, assumption 3.1 holds even in this continuous space.

However, assumption 3.1 is not limited to just finite-sized policy classes. Consider an infinite-sized $\Pi$, where one policy in $\Pi$, called $\pi^*$, always outputs 1, and all other policies are Gaussian policies with $\sigma = 0.0001$ and $\mu$ uniformly sampled from the interval $[0, 0.0001]$. Here, even when $\Pi$ is infinite and contains stochastic policies in a continuous space, assumption B.1 holds.

Although the aforementioned examples are toyish, they demonstrate that although assumption 3.1 is always true when $\Pi$ is finite, this is not the only case. Further characterization for when the assumption holds is left for future work.

### B.2 On the existence of a solution to 4

In our definition of $\Pi^*$ in 2.5 and the formulation of our objective in 2, it is possible that no solution to 2 exists in settings where $\Pi^*$ is empty. This non-existence issue reoccurs in objective 4, where it is possible that no policies exist that achieve $V_{\max}$. In all of these cases, non-existence is a result of an infinite-sized $\Pi$ where a sequence of policies exists in $\Pi$ that come increasingly arbitrarily close to achieving $V_{\max}$ (or, in the case of 2, to achieving the optimal probability of satisfying $\varphi$), without ever reaching the maximum value.

In these cases, we clarify what a policy that optimizes our true objective 5 is actually achieving. Recall that we ultimately aim to optimize a proxy objective of $\mathcal{R}_\pi + \lambda V_\pi$, under a sufficiently large $\lambda$. If the set $\Pi^*$ is empty, then by definition, there does not exist a "gap" between policies in $\Pi^*$ and policies outside of it. In other words, the value of $\epsilon$ is 0, and Assumption 3.1 does not hold. As a result, as $\lambda$ tends to infinity, the sequence of policies that is increasingly optimal with respect to $\mathcal{R}_\pi + \lambda V_\pi$ will always prioritize increasing $V_\pi$ over $\mathcal{R}_\pi$. In other words, if assumption 3.1 does not hold, optimizing 5 will continually try to improve the proxy reward for LTL satisfaction and will ignore resulting changes to the MDP reward $\mathcal{R}_\pi$.

It is theoretically possible that subsequent policies along the aforementioned sequence have arbitrarily different $\mathcal{R}_\pi$, which is undesirable. However, in practice, this does not tend to be the case, and policies that learn to optimize objective 5 exhibit behavior that is apparently both LTL-satisfying and performant in MDP reward, as evidenced by our experiments in Section 5. Generally, values for $\mathcal{R}_\pi$ are not highly unstable along the sequence of policies that are increasingly optimal with respect to $V_\pi$, and allowing $\lambda$ to serve as a hyperparameter effectively enables a tradeoff to value $\mathcal{R}_\pi$ against $V_\pi$ (see Section 5).

## C Proof for Theorem 4.1

We start with some notation. Let $r^\tau_{\mathrm{CyclER}}$ represent the reward function for a trajectory $\tau$ that are returned by the execution of Alg. 1. Let $T(\tau)$ be the set of timesteps when an accepting state in $\mathbb{B}$ is visited for a trajectory. We write the value function for CyclER, letting $\Gamma_t$ be the same function as defined in function 3:

**Assumption C.1.** *Suppose $T_{\max} = \max_{\pi \in \Pi} \mathbb{E}_{\tau \sim \mathcal{M}^P_\pi} \left[ T(\tau) \middle| \tau \not\models \varphi \right] < M$, there is a uniform bound on the last time a bad (non-accepting) trajectory visits an accepting state across all bad trajectories induced by any policy.*

**Lemma C.2.** *Under Assumption C.1, for any $\pi \in \Pi$ and $\epsilon > 0$ we have*

$$|(1 - \gamma)V^{cyc}_\pi - \mathbb{P}[\pi \models \varphi]| \leq \epsilon$$

*when $\gamma \geq (1 - \epsilon)^{\frac{1}{M+1}}$ is chosen appropriately.*

*Proof.* We follow the proof style of Lemma 4.1 from Voloshin et al. (2023). Let $\mathbb{P}[\pi \models \varphi] = p$ be the probability that $\pi$ satisfies the LTL specification $\varphi$. Recall the value function

$$V^{\mathrm{cyc}}_\pi = \mathbb{E}_{\tau \sim \mathcal{M}^\varphi_\pi} \left[ \sum_{t=0}^\infty \Gamma_t r^\tau_{\mathrm{CyclER}}[t] \right]$$

Let $T(\tau)_{(i)}$ be the (random) $i$-th visit to an accepting state in $\mathbb{B}$. Let $T(\tau)_{(0)}, T(\tau)_{(-1)}$ refer to the first and last visit, respectively.

Because $r^\tau_{\mathrm{CyclER}}[t] = \frac{1}{|c|}$ only when a transition in a cycle is taken (and only once per that transition) and the distance between successive visits to an accepting state is $|c|$ then at most $\gamma^i$ reward is accumulated between

successive visits to an accepting state. In other words $\mathbb{E}_{\tau \sim \mathcal{M}_\pi^\varphi}\left[\sum_{t=T_{(i)}+1}^{T_{(i+1)}} \Gamma_t r^\tau_{\text{CyclER}}[t]\right] = \gamma^i$ and therefore $V_\pi^{\text{cyc}} \leq \frac{1}{1-\gamma}$.

Further, every trajectory $\tau$ is decomposable into (1) the partial trajectory up to the first visit (ie. at time $T(\tau)_{(0)}$), the partial trajectory between the first and last visit (ie. between time $T(\tau)_{(0)}$ and $T(\tau)_{(-1)}$), and (3) the remainder of the trajectory. For trajectories that satisfy the LTL specification, $T(\tau)_{(-1)} = \infty$, otherwise $T(\tau)_{(-1)} \leq M$, finite and bounded (by Assumption C.1). For ease of notation, we omit the dependence of $T$ on $\tau$ (i.e. we write $T(\tau)_{(0)}$ as $T_{(0)}$). By linearity of expectation, we can rewrite our previous equation as:

$$V_\pi^{\text{cyc}} = \underset{\tau \sim \mathcal{M}_\pi^\varphi}{\mathbb{E}}\left[\sum_{t=0}^{T_{(0)}} \Gamma_t r^\tau_{\text{CyclER}}[t]\right] + \underset{\tau \sim \mathcal{M}_\pi^\varphi}{\mathbb{E}}\left[\sum_{t=T_{(0)}+1}^{T_{(-1)}} \Gamma_t r^\tau_{\text{CyclER}}[t]\right] + \underset{\tau \sim \mathcal{M}_\pi^\varphi}{\mathbb{E}}\left[\sum_{t=T_{(-1)}+1}^{\infty} \Gamma_t r^\tau_{\text{CyclER}}[t]\right].$$

When a path $\tau$ is accepting, by definition, $\mathbb{E}_{\tau \sim \mathcal{M}_\pi^\varphi}\left[\sum_{t=0}^{T_{(0)}} \Gamma_t r^\tau_{\text{CyclER}}[t]\Big| \tau \models \varphi\right] = 1$ because every accepting initial path will achieve a reward of 1.

By considering accepting trajectories of LTL formula $\varphi$, then $T_{(-1)} = \infty$:

$$V_\pi^{\text{cyc}} \geq \left(1 + \underset{\tau \sim \mathcal{M}_\pi^\varphi}{\mathbb{E}}\left[\sum_{t=T_{(0)}+1}^{\infty} \Gamma_t r^\tau_{\text{CyclER}}[t]\Big| \tau \models \varphi\right]\right) \mathbb{P}[\pi \models \varphi] = \frac{p}{1-\gamma}$$

where the inequality follows from having dropped any value from non-satisfying trajectories. On the other hand, by the law of total expectation, for a lower bound we have:

$$V_\pi^{\text{cyc}} = p \underset{\tau \sim \mathcal{M}_\pi^\varphi}{\mathbb{E}}\left[\sum_{t=0}^{\infty} \Gamma_t r^\tau_{\text{CyclER}}[t]\Big| \tau \models \varphi\right] + (1-p) \underset{\tau \sim \mathcal{M}_\pi^\varphi}{\mathbb{E}}\left[\sum_{t=0}^{\infty} \Gamma_t r^\tau_{\text{CyclER}}[t]\Big| \tau \not\models \varphi\right]$$

$$\leq p\frac{1}{1-\gamma} + (1-p)\frac{1-\gamma^{M+1}}{1-\gamma}$$

where the first term comes from the upper bound on $V_\pi^{\text{cyc}} \leq \frac{1}{1-\gamma}$ and the second term comes from bounding $T_{(0)}$ with a uniform upper bound $M$ by Assumption C.1

Combining the upper and lower bound together and subtracting off $p$ from both sides, we have

$$0 \leq (1-\gamma)V_\pi^{\text{cyc}} - p \leq 1 - \gamma^{M+1}$$

Select $\gamma \geq (1-\epsilon)^{\frac{1}{M+1}}$ which implies that

$$|(1-\gamma)V_\pi^{\text{cyc}} - p| \leq \epsilon$$

□

**Lemma C.3.** *Let $p^* = \max_{\pi \in \Pi} \mathbb{P}[\pi \models \varphi]$. Under Assumption C.1, then any policy $\pi$ optimizing $V_\pi^{cyc}$ (ie. achieving $V_{\max}^{cyc}$} maintains $|V_\pi^{cyc} - p^*| \leq \epsilon$.*

*Proof.* This follows by an identical argument as in Theorem 4.2 in Voloshin et al. (2023), by using Lemma C.2: $V^{\text{cyc}}$-optimizing policy $\pi^*_{\text{cyc}}$ must satisfy $|(1-\gamma)V_{\max}^{\text{cyc}} - p^*| \leq \epsilon$ when $\gamma$ is selected as in Lemma C.2.

□

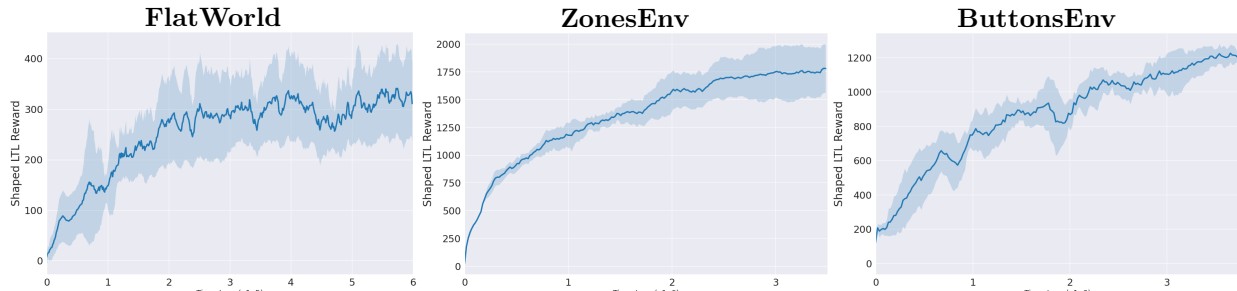

Figure 4: Training curves showing the *shaped* $r_{\text{LTL}}$ achieved by CyclER-learned policies, averaged over 5 random seeds. Each point is the mean of 10 stochastic policy rollouts.

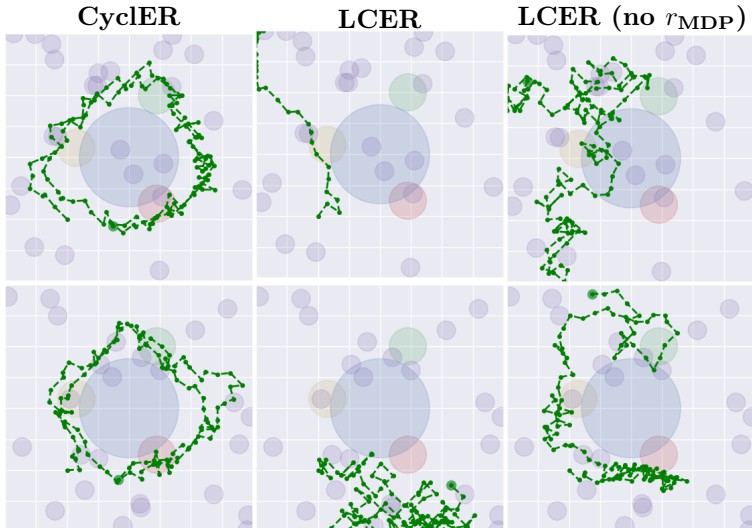

Figure 5: Sample trajectories from each baseline method in the FlatWorld domain (each row is a different seed). CyclER is able to consistently learn behavior that repeatedly visits and accepting state. The LCER baseline cannot achieve this long horizon task and instead optimizes for $r_{\text{MDP}}$. The LCER (no $r_{\text{MDP}}$) baseline, even when successful in visiting the accepting state (bottom right), does qualitatively learn satisfying behavior despite visiting the accepting state.

## D   Additional Experimental Results

**Shaped Reward Training Curves** In Figure 4, we provide training curves for CyclER's performance on the *shaped* LTL proxy reward (i.e., CyclER-shaped reward) in each of our experimental domains. These curves allow us to compare the difference in performance between unshaped and shaped LTL reward and provide insight into a potential correlation between the two. We notice that in all baselines, CyclER-learned policies are able to achieve nonzero shaped reward early in training. Our learned policies achieve nonzero *unshaped reward* (recall the curves in Figure 3) slightly after significant returns in shaped reward are achieved, which supports our hypothesis that CyclER-shaped reward will successfully 'guide' a policy towards visiting the LDBA accepting state. We note that the scales of the CyclER-shaped LTL rewards visualized in Figure 4 vary significantly; the exact numerical values are dependent on specific environments and do not hold any exact meaning with respect to the unshaped LTL reward.

**Qualitative Analysis** To provide more insight into how CyclER learns to repeatedly visit the accepting state, we visualize sample trajectories from policies learned by each of our baselines in the FlatWorld domain, and present these samples in Figure 5. Recall that in this domain, our LTL specification instructs an agent to traverse the red, yellow, and green regions indefinitely, while avoiding the blue region. It is obvious that

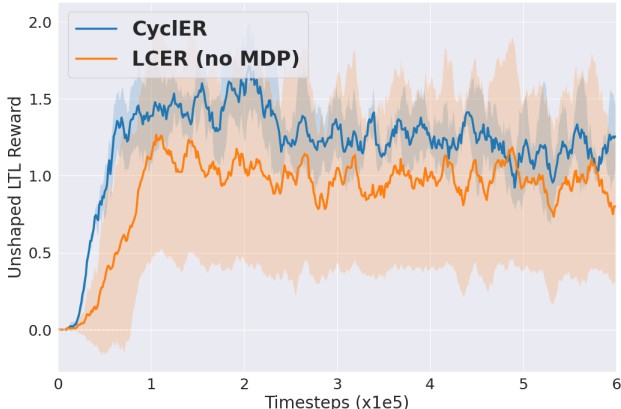

Figure 7: Training curve comparing unshaped LTL reward achieved by CyclER and LCER in the FlatWorld domain with $\varphi = G(F(r)\&F(y))\&G(\neg b)$. Performance is averaged over 5 random seeds. Each point is the mean of 10 stochastic policy rollouts.

the most efficient path to achieve this behavior is to navigate around the blue region and visit each colored region along a circular path.

On the left hand column of Figure 5, we see that policies learned using CyclER are able to perform this behavior, only slightly diverting from the circular path to visit purple regions and collect reward from $r_{\mathrm{MDP}}$. In the middle column, we visualize trajectories from policies learned where LCER is used as $r_{\mathrm{LTL}}$. Due to the sparsity of this reward function, the policy quickly finds areas in the MDP where purple regions are clustered together, and repeatedly visits those areas (in the top middle, it traverses to a corner where the entropy of the policy will affect its position the least.) On the right hand column, we show trajectories from LCER trained without $r_{\mathrm{MDP}}$ in its reward formulation. On the top right, we see that the baseline fails to achieve the task at all. More interestingly, however, on the bottom right, we find that the baseline does indeed achieve the task (i.e., visits the accepting state once), but does not exhibit behavior that qualitatively satisfies our task specification. After completing the task once, the agent turns back in the opposite direction and does not make obvious progress back towards either the red or yellow regions, suggesting that this baseline has not learned to repeatedly satisfy the task.

We also provide sample trajectories from the two policies learned in the experiment addressing question **2** in section 5. The trajectories, visualized in Figure 6, show the difference in behavior between the two successful policies when one is aware of $r_{\mathrm{MDP}}$ during training and when one is not. The MDP reward-aware policy acts notably different, altering its trajectory to reach purple bonus areas (even those that are somewhat far away from the colored regions of interest), while the MDP reward-unaware policy makes no such adjustments and optimizes for completing the LTL task as often as possible.

Figure 6: Trajectories from a CyclER-trained policy on $r_{\mathrm{DUAL}}$ (left) and a CyclER-trained policy on just the shaped $r_{\mathrm{LTL}}$ (right) in the FlatWorld domain.

**Direct comparison in LTL-only setting** To more directly evaluate CyclER's efficacy in alleviating the reward sparsity problem, we recreate the FlatWorld experiment from (Voloshin et al., 2023) with the LTL specification $G(F(\mathrm{red})\&X(F(\mathrm{yellow})))\&G(\neg\mathrm{blue})$. This domain does not have an additional MDP reward to be optimized. As such, the LCER baseline is able to consistently accomplish this objective. This allows us to observe whether CyclER-based reward shaping provides improvements in LTL-guided policy learning even in the absence of an MDP reward. We visualize the training curves in Figure 7. We can see that CyclER not only converges to a satisfying policy more quickly, but consistently achieves a higher reward when compared to the LCER baseline. This confirms our hypothesis that reward shaping is valuable even in settings where the LTL specification serves as the lone objective.

$$\rho(s_{t:t+k}, \top) = \rho_{\max},$$
$$\rho(s_{t:t+k}, f_x(s_t) < c_x) = c_x - f_x(s_t),$$
$$\rho(s_{t:t+k}, \neg\varphi) = -\rho(s_{t:t+k}, \varphi)$$
$$\rho(s_{t:t+k}, \varphi \implies \psi) = \max(-\rho(s_{t:t+k}, \varphi), \rho(s_{t:t+k}, \psi))$$
$$\rho(s_{t:t+k}, \varphi\&\psi) = \min(\rho(s_{t:t+k}, \varphi), \rho(s_{t:t+k}, \psi))$$
$$\rho(s_{t:t+k}, \varphi\|\psi) = \max(-\rho(s_{t:t+k}, \varphi), \rho(s_{t:t+k}, \psi))$$
$$\rho(s_{t:t+k}, G(\varphi)) = \min_{t' \in [t,t+k)} (\rho(s_{t':t+k}, \varphi))$$
$$\rho(s_{t:t+k}, F(\varphi)) = \max_{t' \in [t,t+k)} (\rho(s_{t':t+k}, \varphi))$$
$$\rho(s_{t:t+k}, X(\varphi)) = \rho(s_{t+1:t+k}, \varphi)(k > 0)$$
$$\rho(s_{t:t+k}, (\varphi U \psi)) = \max_{t' \in [t,t+k)} (\min(\rho(s_{t':t+k}, \psi), \min_{t'' \in [t,t')} (\rho(s_{t'':t'}, \varphi))))$$

Figure 9: Quantitative Semantics for LTL.

## E   Quantitative Semantics for LTL

Practitioners have defined *Quantitative Semantics* (QS) for a number of temporal logics in order to quantify the measure of satisfaction for a given specification. These semantics, originally developed to monitor how close hybrid control systems are to violating properties reliant on continuous-value sensor data (Maler & Nickovic, 2004), have since been used in reinforcement learning to learn policies that satisfy formulae in a variety of specification languages, including Signal Temporal Logic (STL) (Li et al., 2017; Balakrishnan & Deshmukh, 2019). However, existing methods of QS for reward shaping fail to effectively extend to indefinite-horizon LTL tasks. In what follows, we will introduce QS for LTL, discuss why naïvely using QS as reward is ill-fitted to deep RL for LTL, and introduce our own approach, extending the CyclER reward shaping method to effectively incorporate QS.

To use QS, we associate each atomic predicate $x \in \text{AP}$ with a *robustness measure* $f_x : \mathcal{S} \to \mathbb{R}$ that quantifies how close $x$ is to being satisfied at state $s$. $x$ evaluates to true at a given state iff $f_x(s) \geq c_x$, where $c_x$ is a constant threshold. We can compose variables in AP with the logical and temporal operators of LTL by introducing QS for each operator, which follows the standard semantics defined for languages like TLTL (Li et al., 2017) and STL (Maler & Nickovic, 2004; Fainekos & Pappas, 2009) and is provided in figure 9. An LTL formula $\varphi$ is true if the quantitative evaluation of $\rho_\varphi > 0$ and false otherwise. We also define a maximum and minimum achievable value for $\rho$ in a given MDP as $\rho_{\max}$ and $\rho_{\min}$, respectively.

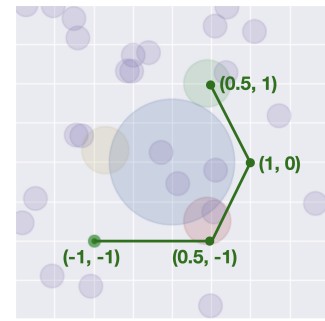

Figure 8: A toy trajectory in the Flatworld MDP.

To better understand quantitative evaluation for a given LTL formula, consider a trajectory (which we will denote as $\xi$) from the Flatworld MDP, shown in figure 8 and the formula $\varphi_{\text{toy}} = F(r)\&G(\neg b)$. We can define robustness measures for our atomic propositional variables as $f_x(s) = \text{distance}(s, x_{\text{center}}) < x_{\text{radius}}$, requiring that the agent be within a region for its variable to evaluate to true.

Let's quantitatively evaluate $\varphi_{\text{toy}}$ on the trajectory $\xi$. For the expression $F(r)$, we find the maximum quantitative evaluation for the variable $r$ in our trajectory. Since our trajectory visits the red region at point (0.5, -1), the evaluation for $r$ at that point is some positive value $c_r > 0$, so the expression $F(r)$ will evaluate to true. For the expression $G(\neg b)$, we negate the quantitative evaluation and find the minimum value for the negated evaluation of $b$. At the point (1, 0), the agent is closest to the blue region, but since it does not enter

it, the minimum value is some positive value $c_b > 0$. The quantitative evaluation for $\varphi_{\text{toy}}$ on $\xi$ is therefore a positive value $\min(c_r, c_b)$, so $\varphi_{\text{toy}}$ evaluates to true for $\xi$.

Suppose we were to naively use the quantitative evaluation of a given LTL specification "off-the-shelf" as a reward function for an RL agent. Since the quantitative evaluation of a trajectory at any given point in time requires evaluating the future states of the trajectory, we can only assign a reward for entire trajectories. For the toy specification considered in our example, the reward would be the smaller of (1) the maximum distance the agent reaches from the blue region and (2) the minimum distance the agent reaches from the red region during the trajectory. Although this seems reasonable as a reward for our example, the quantitative evaluation of an LTL formula becomes increasingly more obscure as the specification increases in complexity. For example, for the specification defined in Figure 1, which instructs the agent to indefinitely oscillate amongst the red, green and yellow regions while avoiding blue, $\xi$ would have a lower quantitative evaluation than a trajectory that does not visit any of the three regions but just barely enters the blue region, even though the latter violates the specification without making any qualitative progress. Moreover, there would be no meaningful difference in quantitative evaluation between $\xi$ and a trajectory that visits red, green and yellow while avoiding blue, or between a trajectory that completes the task twice, or thrice, and so on.

The quantitative evaluation of a trajectory as a reward signal for LTL fails because there are no well-defined terminal conditions for evaluating a finite trace under an infinite-horizon specification (e.g., $\varphi$ from Figure 1), which means that value produced is often useless when used as a reward signal. This is made worse due to the fact that quantitative evaluation of an LTL formula assigns a single value for an entire trajectory, making credit assignment difficult. We propose an alternative reward that avoids these issues, extending the CyclER approach to handle quantitative semantics by rewarding "progress" made by traversing individual transitions in $\mathbb{B}$.

The high-level intuition for our method is as follows: recall that for a given trajectory, CyclER computes hypothetical rewards for each cycle in $\mathbb{B}$, where reward is given if the next transition is taken within that cycle during the trajectory. Our key insight is that we can straightforwardly extend this paradigm to use QS by instead rewarding quantitative progress made towards taking the next transition within an individual cycle. Each transition in $\mathbb{B}$ corresponds to a non-temporal predicate of atomic propositions. Crucially, we can quantitatively evaluate these predicates on individual states, rather than trajectories. When an agent moves to a new state in $\mathcal{M}$, we can quantify how close the agent is to satisfying the transition predicate (and therefore taking the transition in the cycle), and compare it to how close the agent was to satisfying the transition predicate in the previous state. If there is a positive difference between these two values, we reward that progress made towards satisfying the transition predicate.

We present our reward function in function 7 in the main text. In the context of Alg. 1, the reward function defined in 7 will replace $r_{\mathcal{C}}$. Intuitively, we can interpret the reward function defined in 7 as identical to $r_{\mathcal{C}}$, but rewarding quantitative progress towards taking a transition in a cycle rather than only offering reward once that transition is taken. We normalize progress using $\rho_{\max}$ and $\rho_{\min}$ to ensure that the scale of rewards from function 7 remain consistent.

Let us once again return to the example trajectory in Figure 8, with the LTL specification and LDBA from Figure 1 as our objective, and consider the cycle $\{1, 2, 3, 0\}$. We begin in state 1 of $\mathbb{B}$, where the transition we aim to take has the corresponding predicate $r$. When we transition from (-1, -1) to (0.5, -1), we satisfy $r$, and receive positive reward from CyclER because we are closer to $r$ than we were in the previous state. We are now in state 2 of $\mathbb{B}$ and seek to take the transition with predicate $g$. For the transition from (0.5, -1) to (1, 0), we receive positive reward for getting closer to $g$, and for the transition from (1, 0) to (0.5, 1) we receive positive reward for successfully moving closer to $g$. Note that each transition of our trajectory $\xi$ qualitatively makes progress towards visiting the accepting state of $\mathbb{B}$, and this is reflected in the positive rewards assigned by $r_{\text{qs}}$.

In environments where the individual variables in AP are difficult to satisfy, the usage of QS can offer a more dense reward that still leverages the full expressivity of LTL. In our experiments, we show that using the QS version of CyclER strongly outperforms existing approaches of using QS in learning to satisfy indefinite-horizon LTL specifications (Li et al., 2017; Balakrishnan & Deshmukh, 2019) and enables the learning of LTL-compliant policies in complex environments.

---

**Algorithm 2:** Find Minimal Accepting Initial Paths and Cycles (**FindMAIPsAndMACs**)

---

**Input:** LDBA $\mathbb{B}$, accepting states set $\mathcal{B}^*$
Initialize $\mathcal{C}$ to an empty set;
Initialize $\mathcal{P}$ to an empty set;
**DFS**($b_{-1}$, {}, $\mathcal{P}$);
**foreach** *accepting state* $b^* \in \mathcal{B}^*$ **do**
  Initialize *visited* to an empty set;
  Initialize $C$ to an empty set;
  **DFS**($b^*$, {}, $C$);
  Add $C$ to $\mathcal{C}$;
**return** $\mathcal{C}$, $\mathcal{P}$

---

**Algorithm 3: DFS** (Helper for Alg. 2

---

**Input:** Starting node $b$, Path $p$, set $S$
Add node $b$ to *visited*;
**foreach** *Outgoing transition* $(b, \nu, b')$ *from* $b$ **do**
  **if** $b' \in \mathcal{B}^*$ **then**
    Add the transition $(b, \nu, b')$ to $p$;
    Add $p$ to $S$;
  **else**
    **if** $b' \notin visited$ **then**
      Add the transition $(b, \nu, b')$ to $p$;
      **DFS**($b'$, $p$, $S$);
Remove node $b$ from *visited*;

---

# F   Additional Algorithmic Details

**Motivation of visiting frontier.**   To motivate the importance of maintaining the visited frontier $e$ introduced in section 4.2, we show via example that the existence of non-accepting cycles in $\mathbb{B}$ may allow for trajectories that infinitely take transitions in a MAC or MAIP without ever visiting an accepting state.

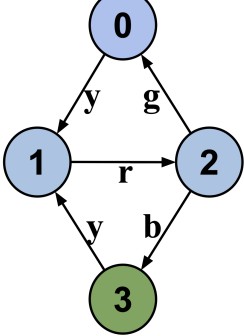

Consider the accepting cycle $\{3, 1, 2\}$ in the partial automaton in Figure 10. Although this cycle is a MAC, there does exist a separate cycle starting and ending at state 1 (i.e. the cycle $\{1, 2, 0\}$.) If we give reward every time a transition in the cycle $\{3, 1, 2\}$ is taken, a policy may be able to collect infinite reward without ever visiting an accepting state. For example, in Figure 10, a path $\{1, 2, 0, 1, 2, 0 \dots\}$ would infinitely take transitions in a MAC, and therefore collect infinite reward without ever visiting the accepting state 3. Our visited frontier $e$ will ensure that rewards will only be given once per transition until an accepting state is visited.

Figure 10: A partial Büchi automaton that necessitates a visited frontier.

## F.1   Finding Minimal Accepting Cycles and Accepting Initial Paths

In algorithms 2 and 3, we include the psuedocode for finding minimal accepting initial paths and minimal accepting cycles in a given $\mathbb{B}$, which constitute the sets $\mathcal{P}$ and $\mathcal{C}$ respectively for usage in algorithm 1.

# G   Additional Experimental Details

**Environments and Tasks.** For each random seed of training, the locations of the following objects were randomized and fixed: in FlatWorld, the location of the bonus areas, in ZonesEnv, the locations of all colored regions and hazard regions, and in ButtonsEnv, the locations of the buttons, bonus areas, and

| Environment | LTL $\varphi$ |
|---|---|
| FlatWorld | $G(F(\text{red})\&X(F(\text{green})\&X(F(\text{yellow}))))\&G(\neg\text{blue})$ |
| ZonesEnv | $G(F(\text{blue})\&F(\text{purple})\&F(\text{red})\&F(\text{green}))$ |
| ButtonsEnv | $G(F(\text{button1})\&F(\text{button2}))\&G(\neg\text{gremlin})$ |

Table 3: Specification for each domain.

gremlins. In ZonesEnv and ButtonsEnv, we use the Point robot from Ji et al. (2023) as our agent, which has a 2-dimensional action space $a \in [-1, 1]^2$.

The observation space and environment used in our ZonesEnv are the the default spaces provided the Zones Level 1 environment in Ji et al. (2023), with the following changes: there are additional lidar observations for each of the four colored zones, and we place four static collidable walls as boundaries to enclose the agent's environments at the border of where objects can be randomly placed. The observation space and environment used in our ButtonsEnv experiments are the default spaces provided the Button Level 1 environment in Ji et al. (2023), with two gremlins and eight bonus regions.

In ZonesEnv and ButtonsEnv, we define a simple robustness measure for each atomic propositional variable in the environments (the red, yellow, green, and purple regions in ZonesEnv, and buttons 1 through 4 and gremlin for ButtonsEnv). The robustness measure for a general variable $x$ at a given state $s$ is defined as follows:

$$f_x(s) = \text{distance}(s, x) \le 0$$

For example, if an agent was a distance of 2 units away from the red region in ZonesEnv, the robustness measure of the variable "red" at that state would evaluate to -2. For ButtonsEnv, where the gremlin variable refers to multiple moving objects, the robustness measure corresponds to the minimum distance from the agent to any gremlin. We set $\rho_{\max} = 0$ in our environments and $\rho_{\min}$ to be the negative largest distance achievable in each environment.

Our LTL task specifications are defined in Table 3. We use the Spot tool Duret-Lutz et al. (2022) to convert our specifications into corresponding LDBAs. We report the number of states, transitions, and *MAC*s in each LDBA used in our experiments in Table 5.

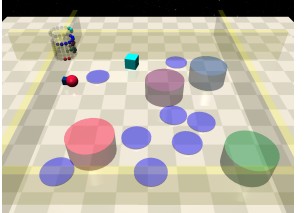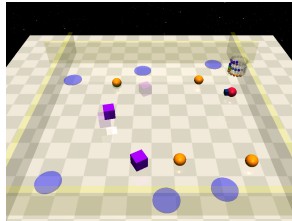

Figure 11: Example visualizations of the ZonesEnv (left) and ButtonsEnv (right) environments.

**Training details.** For all experiments, results are averaged over five random seeds. We provide hyperparameter choices for PPO for each experiment in Table 6 and choices for $\lambda$ in Table 4. In Table 6, batch size refers to the number of trajectories. In our PPO implementation, we use a 3-layer, 64-hidden unit network as the actor using ReLU activations, and a 3-layer, 64-hidden unit network architecture with tanh activations in between layers and no final activation function for the critic. The actor outputs the mean of a Gaussian, the variance for which is learned by a 3-layer, 64-hidden unit network that shares the first 2 layers with the actor policy itself. All experiments were done on an Intel Core i9 processor with 10 cores equipped with an NVIDIA RTX A4500 GPU. We use the Adam optimizer in all experiments.

In Figure 3, reward was computed by evaluating the policy every ten trajectories in the case of FlatWorld, and every 25 trajectories for ZonesEnv and ButtonsEnv. The $r_{\text{LTL}}$ and $r_{\text{MDP}}$ values shown are from averaging performance over ten rollouts for each data point with a smoothing window of size 5.

For the BHNR baseline, we use a partial signal window size of 60 for FlatWorld, 700 for ZonesEnv, and 750 for ButtonsEnv, treating this value as a hyperparameter and performing a sweep to select the window size.

| Environment | CyclER | LCER | TLTL | BHNR |
|---|---|---|---|---|
| FlatWorld | 400 | 1000 | - | - |
| ZonesEnv | 200 | 1000 | 10 | 1 |
| ButtonsEnv | 100 | 1000 | 10 | 1 |

Table 4: Values for $\lambda$ used by baselines for each domain.

| Environment | States | Edges | $MAC$s |
|---|---|---|---|
| FlatWorld | 5 | 18 | 14 |
| ZonesEnv | 8 | 35 | 44 |
| ButtonsEnv | 3 | 9 | 4 |

Table 5: Details for the LDBAs used in each experimental domain.

Our TLTL baseline is trained by assigning the TLTL value of a trajectory as the reward at the end of the trajectory, and using the discounted reward-to-go for each prior timestep as the reward signal. For our TLTL and BHNR baselines, we tried computing $r_{\text{LTL}}$ in two ways: first, we tried using the original quantitative evaluation of the formula as the reward, where the resulting rewards were mostly negative due to how we defined our robustness measures for each variable. To evaluate if the sign and magnitude of the reward caused difficulty during learning, we normalized the quantitative evaluation done in TLTL and BHNR using $\rho_{\text{min}}$ and $\rho_{\text{max}}$, so that the reward value would be between 0 and 1. We found that this adjustment did not have a significant impact on either baseline's ability to learn $r_{\text{LTL}}$, but it did allow for easier optimization of $r_{\text{MDP}}$; therefore, we report the results from the normalized evaluation in our experiments.

| Environment | Critic LR | Actor LR | $\alpha$ | Update freq. | $\gamma$ | Batch size | $|\tau|$ (training) |
|---|---|---|---|---|---|---|---|
| FlatWorld | 0.001 | 0.0003 | - | 1 | 0.98 | 128 | 120 |
| ZonesEnv | 0.0125 | 0.0025 | 0.3 | 3 | 0.99 | 128 | 700 |
| ButtonsEnv | 0.001 | 0.0003 | 0.2 | 3 | 0.99 | 128 | 750 |

Table 6: Hyperparameters used during training for each domain.

