# OpenReview forum: "LTL-Constrained Policy Optimization with Cycle Experience Replay"
_TMLR — Accepted by TMLR_

### Review · Reviewer_AdC7 · 2024-11-21

**Summary Of Contributions:**

The paper proposes a new reward-shaping approach for model-free RL with LTL task specification. Specifically, the authors propose a reward function with an LTL and an MDP component. The LTL component is usually a sparse reward for visiting accepting states. This paper proposes a reshaping approach to make the LTL reward more dense. The authors evaluate their approach and existing baselines on multiple benchmarks and observe that their approach performs better with respect to the LTL objective than approaches with sparse rewards.

The clarity of the paper varies between sections as well as the writing quality.

**Audience:**

Yes

**Claims And Evidence:**

Yes

**Requested Changes:**

- The abstract needs improvement. The problem addressed and contributions are not clear enough, and the language can be improved.
- The introduction is not well written and rather confusing:
   * the work is not well motivated by the first paragraph,
   * the second paragraph requires references
   * in the third paragraph it is unclear what RL-based reward optimization means and it is not clear what the novelty of the proposed approach is
    *  (+) the last sentence in the third paragraph is too strong or too vague: there is work that uses temporal logic (TL) for constraining and deep RL for performance objectives [1], there is work that uses TL for deep RL where the specification describes [2-4]
   * it seems that there are three "in this paper paragraphs" - please restructure to one concise paragraph
   * for the fourth paragraph the fit into the overall story of the introduction seems off
- Fig. 1 requires references after "unlike other approaches"
- The contributions are not concise enough. This work is not the first on deep RL and TL (this statement is repeated multiple times - please revise everywhere), see previous comment marked with (+)
- Can you discuss the scalability of your reward shaping with an increasing number of states?
- If transitions are scarce, then the reshaped reward in Eq 6 or 7 is still very sparse. Can you add to your experiments how many transitions are in the automata? And can you discuss this issue?
- The first two paragraphs in Sec. 4.3. distract from the proposed work. In my opinion, they are either preliminaries or related work.
- The experiments are solid, yet they are not described and discussed well enough:
  * it would be good to report the shaped rewards for the first set of experiments. Right now it is not clear how much information gain is accomplished with the new reward
  * it would be good to introduce a notation that clarifies when the quantitative $r_{LTL}$ reward is used and when the transition-based shaped reward.
  * The question (2) is not well demonstrated and discussed. If you do not want to make a table for the results, then rather state relative metrics between ablations. Also, what do you observe qualitatively?
  * the usage of $r_{LTL}$ for unshaped and shaped reward is confusing
- Fig 3: It is unclear what the x-label "trajectories" means and for how many training steps the approaches are trained with PPO.
- Table 1: I think $r_{LTL}$ is confusing since this is the unshaped reward. Reporting satisfaction cycles or similar would be much better.
- The literature review in the appendix sometimes repeats the exact same sentences and it is unclear what the criteria are for the literature presented in the main text and the literature in the appendix. I would suggest having a concise but full literature review in the main text.

Minor:
- The notation on page 3 for $b_0$ seems inconsistent. Or what is $b_{-}$?
- In Sec. 4 your proposed approach becomes clear. Yet, it would be better if you could achieve this in the introduction already.
- I believe the "column" formatting of Alg. 1 and Tab. 2 is not aligned with style requirements. I think these should be adapted to single-column format. Additionally, the tables could be made more visually appealing.
- Can you relate/discuss your work with respect to safe RL (e.g., [5-8]) where safety (in your case the task specification) and performance objectives (in your case the MDP reward) are also often incorporated differently?

References:

[1] Safe Reinforcement Learning with Probabilistic Guarantees Satisfying Temporal Logic Specifications in Continuous Action Spaces. Hanna Krasowski, Prithvi Akella, Aaron D. Ames, and Matthias Althoff. In Proc. of the IEEE Conference on Decision and Control (CDC), 2023

[2] A. Lavaei, F. Somenzi, S. Soudjani, A. Trivedi and M. Zamani, "Formal controller synthesis for continuous-space MDPs via model-free reinforcement learning", Proc. of the ACMIIEEE International Conference on Cyber-Physical Systems, pp. 98-107, 2020.

[3] D. Gundana and H. Kress-Gazit, "Event-based signal temporal logic synthesis for single and multi-robot tasks", IEEE Robotics and Automation Letters, vol. 6, no. 2, pp. 3687-3694, 2021.

[4] M. Cai, M. Hasanbeig, S. Xiao, A. Abate and Z. Kan, "Modular deep reinforcement learning for continuous motion planning with temporal logic", IEEE Robotics and Automation Letters, vol. 6, no. 4, pp. 7973-7980, 2021.

[5] Hanna Krasowski, Jakob Thumm, Marlon Müller, Lukas Schäfer, Xiao Wang, and Matthias Althoff, Provably Safe Reinforcement Learning: Conceptual Analysis, Survey, and Benchmarking, Transactions on Machine Learning Research, 2023

[6]  Mohammed Alshiekh, Roderick Bloem, Rüdiger Ehlers, Bettina Könighofer, Scott Niekum, and Ufuk Topcu.
Safe reinforcement learning via shielding. In Proc. of the AAAI Conf. on Artificial Intelligence (AAAI),
pp. 2669–2678, 2018.

[7] Xiao Li, Zachary Serlin, Guang Yang, and Calin Belta. A formal methods approach to interpretable reinforcement learning for robotic planning. Science Robotics, 4(37), 2019a

[8] Bettina Könighofer, Florian Lorber, Nils Jansen, and Roderick Bloem. Shield synthesis for reinforcement
learning. In Leveraging Applications of Formal Methods, Verification and Validation: Verification Princi-
ples, pp. 290–306, 2020.

**Strengths And Weaknesses:**

Strengths:
- Innovative idea for the problem that LTL rewards are usually sparse and might not include all objectives that should be considered
- Solid evaluation with different ablations on three environments.

Weaknesses:
- The proposed concept is not clearly presented in the Abstract and Introduction
- The contributions are formulated too vaguely throughout the text and have to be more concise

---

> ### Author Response · Authors · 2024-12-17
> **Author response to reviewer AdC7**
>
> We thank the reviewer for their very thorough review and constructive comments and suggestions! We address the reviewers comments and suggestions in turn:
>
> > The abstract needs improvement. The problem addressed and contributions are not clear enough, and the language can be improved.
>
> We have revised the abstract to more clearly delineate the problem motivation, which is the sparsity issue of LTL, and our contributions, which is the CyclER algorithm and its accompanying theory and experimental results.
>
> > The introduction is not well written and rather confusing.
>
> We have revised the introduction to incorporate the following changes:
> * We have restructured the third and fourth paragraphs to more clearly outline the problem we are considering and the main challenge associated with it (reward sparsity.)
> * We thank the reviewer for bringing these related works to our attention. We would like to point out that [1] and [3] consider STL, not LTL, and [2] considers a fragment of LTL. [4] is focused on LTL satisfaction without the presence of additional rewards. Nonetheless, we have added these citations to our introduction and more clearly distinguished our problem’s novelty: our approach considers general LTL specifications that constrain reward optimization and enables deep RL approaches for continuous state and action spaces.
> * We have relaxed the language in the introduction to make the claims less vague and more precise.
>
> > Fig. 1 requires references after "unlike other approaches".
>
> We have added references to approaches that support this claim.
>
> > The contributions are not concise enough. This work is not the first on deep RL and TL (this statement is repeated multiple times - please revise everywhere), see previous comment marked with (+)
>
> We have tightened the structure of the introduction to more quickly introduce the problem and its challenge.
>
> > Can you discuss the scalability of your reward shaping with an increasing number of states?
>
> The computational performance and effectiveness of CyclER will vary as the number of accepting cycles in an LDBA increases, which is often correlated with but is not directly related to the number of states in an LDBA. For example, an LDBA could have 100 states but just one accepting cycle that loops through all states. However, it is generally true that the number of accepting cycles correlates with the number of states in an LDBA. As such, if the number of states (and therefore accepting cycles) increases, CyclER will incur additional computational overhead (1) when computing all cycles prior to policy learning and (2) when keeping track of all potential reward values for each cycle for each trajectory stored in an agent’s replay buffer. However, if the increasing number of states leads to cycles that are on average longer in length, we anticipate that the algorithmic performance of CyclER in relation to unshaped LTL reward will improve. Longer cycles will allow for more transitions to receive reward, further alleviating the sparsity issue. We provide a brief discussion of the computational overhead of CyclER in the third paragraph Section A of our appendix.
>
> > If transitions are scarce, then the reshaped reward in Eq 6 or 7 is still very sparse. Can you add to your experiments how many transitions are in the automata? And can you discuss this issue?
>
> It is true that with very few transitions in the LDBA, Eq. 6 will provide little advantage to existing unshaped reward methods for LTL satisfaction. However, the same is not true for Eq. 7. Even with just one transition, if the predicate associated with that transition has an informative quantitative semantics, CyclER-shaped reward will receive nonzero reward at every timestep. We provide a discussion regarding this in the first paragraph of Section A in our appendix. To further clarify this, we added a sentence in section 4.3 that explains that quantitative semantics can lead to dense LTL reward even with few transitions in our LDBA. We have also added to our experimental details section (appendix G) how many transitions and states are in each automata.
>
> > The question (2) is not well demonstrated and discussed. If you do not want to make a table for the results, then rather state relative metrics between ablations. Also, what do you observe qualitatively?
>
> We have added a qualitative analysis comparing a reward-aware CyclER approach to one that is not reward-aware in our additional experimental results section (appendix D).

---

> > ### Author Response · Authors · 2024-12-17
> > **Author response to reviewer AdC7 (continued)**
> >
> > > the usage of $r_\text{LTL}$ for unshaped and shaped reward is confusing.
> >
> > In our experiment results, we only use $r_\text{LTL}$ to refer to the **unshaped** reward (i.e. the number of accepting state visits achieved in a trajectory), for the sake of consistent reporting with our baselines and to avoid confusion with shaped reward usage. We have modified the language in section 5 to more carefully avoid any confusion here, and modified the plots in Figure 3 to clarify that the unshaped LTL reward is reported. If there are lingering concerns, please let us know and we are happy to make further adjustments.
> >
> > > Fig 3: It is unclear what the x-label "trajectories" means and for how many training steps the approaches are trained with PPO.
> >
> > The “trajectories” label refers to the number of fixed-horizon, finite-length trajectories that the policy is trained on. We have changed these axes in Fig. 3 to instead report the number of training steps the policy is trained on.
> >
> > > Table 1: I think $r_\text{LTL}$ is confusing since this is the unshaped reward. Reporting satisfaction cycles or similar would be much better.
> >
> > We have changed the column title in Table 1 from $r_\text{LTL}$ to “Buchi accepting state visits” to clarify this confusion.
> >
> > > The literature review in the appendix sometimes repeats the exact same sentences and it is unclear what the criteria are for the literature presented in the main text and the literature in the appendix. I would suggest having a concise but full literature review in the main text.
> >
> > We have adjusted our related works section in the main text to concisely refer to the same body of previous work as is addressed in the appendix’s extended review, and have removed the extended review from the appendix.
> >
> > > Can you relate/discuss your work with respect to safe RL (e.g., [5-8]) where safety (in your case the task specification) and performance objectives (in your case the MDP reward) are also often incorporated differently?
> >
> > We thank the reviewer for bringing these works to our attention. Our work differs from the set of papers surveyed in [5], where methods are provided to correct potentially already-learned unsafe policies from exhibiting unsafe behavior. In our work, “unsafe” can be thought of as policies that do not satisfy our LTL task. However, our work differs in that we provide a learning-based approach, rather than a means of verifying or correcting an existing policy with respect to a safety specification. The nature of our provided guarantee is also different: we provide a guarantee that the optimal policy with respect to our objective will assuredly accomplish the LTL task with optimal probability. Similarly, [6] and [8] provide ‘shielding’ techniques, where safety is assured by learning a policy that switches between a performant but potentially unsafe policy and a safe but potentially less performant policy. Our work learns only one policy that aims to accomplish a formal task specification alongside an optimization objective. [7] uses a specific type of temporal logic, called TLTL, to guide a reinforcement-learning controller to accomplish robotic tasks. We directly compare against TLTL in our experiments.
> >
> > Minor: We have fixed the notation in page 3 for $b_0$.
> >
> > We would like to sincerely thank the reviewer for the suggestions and feedback, which we have incorporated into our updated manuscript. We feel that this has significantly strengthened the quality and clarity of our paper. If the reviewer has additional questions, we are happy to address them.

---

> ### Comment · Reviewer_AdC7 · 2024-12-23
> **Response to rebuttal**
>
> Dear authors,
>
> Thanks a lot for your revisions and rebuttal. Especially, your additional explanations for the QS semantics are very helpful.
>
> - The abstract still does not clearly state the situation and problem. In particular, the first three sentences are not clear: why is LTL insufficient for task specifications? If so, it does make little sense that you use LTL? Isn't the problem rather that current methods using LTL for constraining RL are only sparsly exposing constraints? Why is this constraint optimization difficult in RL and not in others?
> - The second paragraph in the introduction still doesn't have any references. But contains phrases like "but few works consider both rewards and specifications in the same setting" that require references.
> - When using quantitative semantics, you seem to regard signals to compute them. Thus, this is a sublanguage of STL. As a consequence, I would expect that you discuss the difference of your QS approach to STL approaches in your related work section. The current version does not make the differences clear and simply states that they are different languages.
> - As of now, I would disagree with this statement "To the best of our knowledge, this work is the first to extend policy optimization under general LTL constraints to continuous spaces (and thereby DRL), " (Sec. 6)  since there are works with STL [1-2], which is a more general language. I think your main contribution is the reward shaping, and thus, this statement is too broad.
> - Some of my comments about the experiments have not been addressed or answered. I would appreciate a short explanation, e.g., why not report the shaped reward for CyclER at least in the appendix? Why not add some highlighting to Tab. 1 and Fig. 2 that clarifies when Eq. 6 and when Eq. 7 is used ( I know you state this, but this is a major setup difference that should be clear in the exhibits, in my opinion).
> - Now that you changed the header of Tab 1, I am wondering if changing the y-axis for the upper plots in Fig. 3 to B* visits would be better as well since only some of the approaches reported actually use the unshaped reward during training. This comment also relates to my previous question: although it is not directly comparable, it would be good to report the "actual" rewards the RL agents obtain somewhere so the reader gets an intuition about the empirical change of the reward signal due to your shaping approach.
> - Thanks for your discussion on the relation to safe RL. I would appreciate
>
> [1] P. Varnai and D. V. Dimarogonas, "On Robustness Metrics for Learning STL Tasks," 2020 American Control Conference (ACC), Denver, CO, USA, 2020, pp. 5394-5399, doi: 10.23919/ACC45564.2020.9147692.
> [2] Singh, N. K., & Saha, I. (2023). STL-Based Synthesis of Feedback Controllers Using Reinforcement Learning. Proceedings of the AAAI Conference on Artificial Intelligence, 37(12), 15118-15126. https://doi.org/10.1609/aaai.v37i12.26764

---

> > ### Author Response · Authors · 2024-12-25
> > **Author response to AdC7**
> >
> > Thanks to the reviewer for their continued engagement with our work and for their helpful feedback! We have updated our manuscript to incorporate the reviewer's suggestions, and address each point in turn:
> >
> > > The abstract still does not clearly state the situation and problem. In particular, the first three sentences are not clear: why is LTL insufficient for task specifications? If so, it does make little sense that you use LTL? Isn't the problem rather that current methods using LTL for constraining RL are only sparsly exposing constraints? Why is this constraint optimization difficult in RL and not in others?
> >
> > We say that LTL is insufficient because sometimes we have optimality conditions (MDP reward) alongside a satisfaction condition (LTL, or a task that should be formally specified.) We agree that the original wording in the abstract, where we say “LTL is insufficient for task specifications” is a bit unclear, and have adjusted the wording in the abstract to more clearly specify this by saying “LTL is insufficient to capture both [satisfaction and optimality conditions]”.
> >
> > We agree that the primary problem we address is that LTL proxy rewards are sparse. However, the motivation for this sparsity comes exactly from the LTL-constrained optimization setting. In settings where the LTL specification is the lone objective, exploration is sufficient to find a satisfactory policy via RL. But, in LTL-constrained optimization settings where a separate MDP reward is present, this is not nearly as much the case: a policy can get ‘distracted’ by the MDP reward during learning due to the sparsity of existing LTL proxy rewards. This is our main motivation for solving the sparsity issue by introducing CyclER.
> >
> > > The second paragraph in the introduction still doesn't have any references.
> >
> > We’ve added a reference to [1] in the second paragraph which considers an identical setting (satisfying general LTL specifications alongside optimizing general MDP rewards.)
> >
> > [1] Policy Optimization with Linear Temporal Logic Constraints. Voloshin et. al. NeurIPS 2022.
> >
> > > When using quantitative semantics, you seem to regard signals to compute them... As a consequence, I would expect that you discuss the difference of your QS approach to STL approaches in your related work section.
> >
> > We agree - we provide the distinction between our work and existing works’ usage of STL explicitly in section 4.3 and appendix E, but have added language to our related works section to reflect this distinction and refer to the aforementioned sections to give full context to the reader.
> >
> > > As of now, I would disagree with this statement "To the best of our knowledge, ... " (Sec. 6) since there are works with STL [1-2], which is a more general language. I think your main contribution is the reward shaping, and thus, this statement is too broad.
> >
> > We agree that the reward shaping is the main contribution, and we’ve changed the wording to no longer make as broad of a statement.
> >
> > > I would appreciate a short explanation, e.g., why not report the shaped reward for CyclER at least in the appendix? Why not add some highlighting to Tab. 1 and Fig. 2 that clarifies when Eq. 6 and when Eq. 7 is used ( I know you state this, but this is a major setup difference that should be clear in the exhibits, in my opinion).
> >
> > We felt originally that reporting the shaped reward for CyclER could potentially be misleading to a reader, since the reward is often at a very different scale (and does not always exactly correlate with the unshaped reward.) We did not want to show multiple different reward plots to a reader while they were trying to understand the meaning of our general conclusions in the first place. However, we agree that this would be a valuable addition to the appendix, and have added these shaped reward curves in Figure 4 along with commentary in Appendix D.
> >
> > Yes, the difference in using eq. 6 and 7 is a substantial setup difference that we do want to be clear about. We discuss when eq. 6 and 7 are used in text mostly for the sake of visual consistency in our figures. Highlighting the difference in setup may end up being misleading to a reader - we want to emphasize that the approaches are more similar than different since the same core CyclER approach is being used in both cases (eq. 6 or 7.) We also want to avoid creating potential inconsistencies across figures. For example, we could create an entirely new column in table 1, but this would be inconsistent with the graphs, since there is no separate curve for the QS version of CyclER, so it would appear that one fewer baseline is missing. We have made some adjustments to further emphasize this difference in other ways like bolding the relevant text in section 5.2.

---

> > > ### Author Response · Authors · 2024-12-25
> > > **Author response to AdC7 (continued)**
> > >
> > > > Now that you changed the header of Tab 1, I am wondering if changing the y-axis for the upper plots in Fig. 3 to B* visits would be better ..., it would be good to report the "actual" rewards the RL agents obtain somewhere so the reader gets an intuition about the empirical change of the reward signal due to your shaping approach.
> > >
> > > We do want to be clear that the y-axis in the plots for Fig. 3 (upper) is not exactly the same as B* visits. This axis reports a discounted reward (using the eventual discounting reward proxy in eq. 3) which is what is used to optimize the policy during training. This is different from Table 1, where we exactly report B* visits. We mistakenly conflate the two in the text so we’ve removed this conflation to avoid confusion.
> > >
> > > Regarding the “actual” (shaped) rewards - we have added this to the appendix, as mentioned in our previous response.
> > >
> > > We thank the reviewer again for their continued discussion, helpful suggestions, and thoughtful responses. We have incorporated this feedback into our revised manuscript and do feel that the work is significantly more clear as a result. Please let us know if you have further questions and we are happy to further engage in discussion.

---

> ### Comment · Reviewer_AdC7 · 2024-12-30
> **Response to second rebuttal**
>
> Thanks a lot for your detailed response.
>
> The abstract is significantly clearer now and conveys the contributions of the paper.
>
> "In contrast, our work allows for policy optimization under general LTL constraints, which are often more sparse than signals provided by STL, and extends this paradigm to continuous spaces (and thereby DRL)."
>
> Is this the sentence you added for LTL vs STL? This does not discuss the difference to STL of your work in my opinion. To clarify, in Figure 9 it seems that the quantitative evaluation is based on a signal, e.g., c - f(s_t). Thus, it seems to me that your used LTL with QS is a subset of STL. Can you point me to where you clarify this or make the distinction in Sec. 4.3 and Appendix E. As of now, I think you only state that STL is different from LTL (which is trivially given by the different semantics) but not why this is a better fit for your approach. Also, I think the statement in the brackets is incorrect since there is also DRL in discrete spaces and it can leave the impression that your approach makes DRL with LTL constraints feasible, while this is also possible/done with other existing approaches.
>
> Thanks for adding the plots to Appendix D. One suggestion you might want to consider is adapting the plots by adding the unshaped rewards and normalizing the shaped rewards with the final mean value. I know this distorts the shaped reward curve, but it could intuitively the information gain of the shaped reward and allow for a more direct comparison.
>
> Thanks also for the other adaptions.

---

> > ### Author Response · Authors · 2025-01-01
> > **Author response to Reviewer AdC7**
> >
> > Thanks to the reviewer for their continued engagement and discussion.
> >
> > >  This does not discuss the difference to STL of your work in my opinion. To clarify, in Figure 9 it seems that the quantitative evaluation is based on a signal, e.g., c - f(s_t). Thus, it seems to me that your used LTL with QS is a subset of STL... the statement in the brackets is incorrect since there is also DRL in discrete spaces.
> >
> > It is true that LTL with QS is a subset of STL. We only implicitly make the distinction between STL and LTL-QS in the third paragraph of Section 6 and in paragraphs 1, 5, and 6 of Appendix E, and we agree that being more explicit will be helpful. We have accordingly revised the sentence referenced by the reviewer to "Our CyclER-QS method can be viewed as an approach designed for a subset of STL, with improved performance by our leveraging of the LDBA during policy learning." We thank the reviewer for their help in making our contribution more clear.
> >
> > >  One suggestion you might want to consider is adapting the plots by adding the unshaped rewards and normalizing the shaped rewards with the final mean value. I know this distorts the shaped reward curve, but it could intuitively the information gain of the shaped reward and allow for a more direct comparison.
> >
> > This is an interesting potential visualization - however, we do think that since this value was not used in the actual policy learning step (unlike the plots added in our most recent revision), this could be a bit more obscuring to the reader, especially if the reader starts to contrast it with the plots that more accurately reflect the signal given to the policy during learning. If the reviewer has any further suggestions we're happy to continue the discussion.
> >
> > Thanks again! And please let us know if this helps clarify things, and if you have any further questions. We look forward to any continued discussion with the reviewer.

---

> ### Comment · Reviewer_AdC7 · 2025-01-02
> **Response to third rebuttal**
>
> Thanks a lot for addressing my questions and comments. The contributions are now much clearer than in the initial submission and the claims are sufficiently backed with evidence. Thus, I adjusted my overall assessment.

---

### Review · Reviewer_srmk · 2024-11-24

**Summary Of Contributions:**

This paper introduces CyclER, a method for Deep Reinforcement Learning with continuous state and action spaces under general Linear Temporal Logic constraints. The key idea is to post-hoc reward-shape past trajectories in the MDP (thus, Experience Replay) using a metric of partial progress towards either a first accepting state in the Büchi automaton corresponding to the LTL objective, or a cycle since the last state, with higher rewards given to shortest paths / cycles, and only once per transition. The authors then optimize for a Lagrangian objective that combines the MDP's reward and the LTL objective, and propose a method to incorporate Quantitative Semantics into CyclER. In experiments in 3 environments, CyclER finds policies that effectively satisfy the LTL constraint, while still seeking the MDP's rewards.

**Audience:**

Yes

**Claims And Evidence:**

Yes

**Requested Changes:**

(Not critical, but I think would significantly strengthen the experimental argument): Compare to LCER in environments where LCER was initially successful on, to isolate just the influence of reward shaping in satisfying LTL. It should suffice to just run on the 2 variations of FlatWorld (no MDP rewards) that LCER used. This would be a cleaner comparison to justify that the reward shaping alone is helpful, even in the absense of the MDP rewards.

**Strengths And Weaknesses:**

# Strengths

* The paper is generally well written and polished.
* The method and contributions are well motivated, and fills in a gap in the RL+LTL literature in providing a method that (1) is based on deep RL (allowing continuous MDPs), (2) combines both RL and LTL objectives.
* The main contribution, the reward shaping method itself, CyclER, is quite intuitive, and seems to produce much smoother rewards than prior work (notably LCER).

# Weaknesses

I think the main weakness is in the experimental evaluation. The authors show 3 MDPs where CyclER effectively optimizes $r_{LTL}$, while the baselines essentially fail (e.g., even LCER with the LTL reward only, though it has non-zero performance). This essentially shows, by an existential argument, that there are situations where "CyclER works", whereas prior work doesn't. I generally believed this when reading the paper, since the method seems sensible to me. Thus, it would have been more informative to understand where the boundary lies, or how far can CyclER go. Given that LTL has an inherent sparseness/long-horizon challenge to it, it would have been very helpful to get a sense of how sparse can the LTL reward be such that CyclER can still satisfy it within a reasonable computational budget, and where that breakdown point is compared to LCER or other baselines.

---

> ### Author Response · Authors · 2024-12-17
>
> We thank the reviewer for their thoughtful review and constructive feedback! We address the reviewers comments and suggestions in turn:
>
> >  Thus, it would have been more informative to understand where the boundary lies, or how far can CyclER go. Given that LTL has an inherent sparseness/long-horizon challenge to it, it would have been very helpful to get a sense of how sparse can the LTL reward be such that CyclER can still satisfy it within a reasonable computational budget, and where that breakdown point is compared to LCER or other baselines.
>
> We agree with the reviewer that understanding the exact level of sparsity that CyclER is able to handle is valuable and will help better characterize our reward shaping technique. Ultimately, the success of CyclER is limited to the quality and achievability of the atomic propositional variables (APs) present in a given environment. If there are very few APs in an LTL formula (and therefore an LDBA with very few states and transitions), there will be fewer cycles that CyclER can leverage to better shape reward. This shortcoming can be overcome with the use of quantitative semantics (QS, as described in section 4.3 and appendix E), but will again be dependent on the quality of the robustness functions associated with each AP. We provide a discussion of these points in our Limitations section (appendix A), but are happy to expand upon it. Ultimately, finding the boundary of CyclER's efficacy is a complex question that relates to the size of the LTL formula (and corresponding LDBA), the achievability of the APs, and the difficulty of the environment itself. We see the exploration of this question as a compelling direction for future work, but one that is out of scope of our current work.
>
> > (Not critical, but I think would significantly strengthen the experimental argument): Compare to LCER in environments where LCER was initially successful on, to isolate just the influence of reward shaping in satisfying LTL. It should suffice to just run on the 2 variations of FlatWorld (no MDP rewards) that LCER used. This would be a cleaner comparison to justify that the reward shaping alone is helpful, even in the absense of the MDP rewards.
>
> We thank the reviewer for their helpful suggestion! We have updated the manuscript to provide an additional experiment in appendix D where we compare against LCER in the Flatworld environment for specification *G(F(r) & F(y)) & G(~b)*, which is the same domain and task used in the original LCER paper. Our results show that CyclER improves on LCER in both convergence speed and quality of the resulting learned policy. Regarding the other FlatWorld task from the original LCER paper, *F(G(y))*, we note that this specification in particular will not provide any additional reward shaping when using CyclER v. LCER due to the structure of the LDBA, which only has one accepting cycle that loops the accepting state back to itself. We note that such specifications do not allow for additional reward shaping from CyclER (without the use of QS) in our limitations section. We have updated our manuscript to reflect these changes.
>
> We again thank the reviewer for their feedback, which we feel has greatly strengthened the experimental argument and quality of the paper.

---

### Review · Reviewer_tqsv · 2025-01-27

**Summary Of Contributions:**

The paper investigates a constrained RL problem, where the objective is to find a policy maximising a cost objective amongst the policies satisfying a high-level logical specification provided in linear temporal logic (LTL) with maximal probability.
It is shown that the constrained problem can be transformed to an unconstrained one via a Lagrange multiplier and eventual discounting provided certain hyperparameters are chosen sufficiently high.
Thereafter, a denser reward signal for LTL satisfaction is provided, which acknowledges partial progress to accepting states and re-distributes the reward at accepting states.
Leveraging quantitative semantics, a refined reward signal is presented, which is denser for LDBAs with few transitions.
Finally, the benefits are demonstrated empirically.

**Audience:**

Yes

**Claims And Evidence:**

Yes

**Requested Changes:**

It is very important to improve the presentation of Sec. 4. In particular, I would appreciate a full worked example for CyclER and a better explanation of Fig 1 (the label is misleading: there is no actual formal MDP and the part on the very right isn’t really explained).
It would be helpful to have the experiments available online to independently verify them.

Minor points:
- p. 3 "An LDBA _may_ have separate": by _definition_ an LDBA always has those separate components
- p.3 "not including the initial state $b_{-1}$": why is $b_{-1}$ not included?
- p.4 make it clear in notation that $j$ depends on $t$, e.g. use $j_t$
- Theorem 3.2: consistently use roman font for $R_{max/min}$
- Def. 4.1: check $b_0$ vs. $b_{-1}$
- p. 6 what is meant by “complete trajactory”
- Eq. 6: subscript $b_t$ missing on RHS
- P.6 “once per visit to accepting state”: Obscures that reward is also provided at non-accepting states
- Theorem 4.1: Formally state Theorem 4.1 in appendix and clarify relation to Lemma C.3
- Algorithm 1: It would aid understanding to reset e explicitly
- Eq. 7: check index $b_{t+1}$
- P. 9 “visits value” not defined

- p. 17: definition of $T^M(s,a,s')$ isn't given
- p. 17: last paragraph is unclear
- p. 18: Can you give a counter-example for the emptyness of $\Pi^*$? Can this be fixed by compact action spaces?
- App. B.2 middle pargraph: What does it mean for $\lambda$ to be "sufficiently large"? Doesn't it need to be infinite in that case
- p. 22 last line typo in coordinate

**Strengths And Weaknesses:**

# Strengths
Whilst this problem seems to have been studied in only one prior paper [Voloshin et al. 2022], it is natural and relevant.
The approach to reward shaping is innovative, reasonably intuitive and shows promise in the experiments, which are presented in good detail.

# Weaknesses
There is a significant lack of clarity in Sec. 4. There is a lot of semi-formal prose and it feels like a lot of clarifications have been moved to the appendix in such a way that just reading the main text is insufficient for grasping the ideas.

I think the paper might have benefitted from focussing on the simpler problem of RL where the single objective is to maximise LTL objectives. The constrained setting mostly adds orthogonal challenges (although reward sparsity becomes an even bigger problem). That way, the presentation could be improved and space is created to explain the main ideas CyclER with QS in sufficient detail in the main body.

The environments seem pretty simple and similar. There is no empirical comparison to other works for the RL + LTL special case apart from Voloshin et al. 2023.

# Questions
- What is the intuition why re-distribution of rewards in CyclER help? I guess it depends on how advantage is estimated in PPO? Is bootstrapping used?
- It is not clear how CyclER with QS can provide a dense reward signal. E.g. consider a reachability task which requires lots of transitions in the MDP but only self-loops in the initial state of the LDBA are taken. Then it seems to me that your method only provides small rewards twice: the first time the loop is taken and when the accepting state is reached. On the other hand, a large number of transitions in the MDP, which make slow but steady progress, remain unrewarded. Am I missing something?
- Are there theoretical guarantees for CyclER with quantitive semantics?

---

> ### Author Response · Authors · 2025-02-01
> **Author response to reviewer tqsv**
>
> We thank the reviewer for their thoughtful review and constructive feedback! We address the reviewers comments and suggestions in turn:
>
> > It is very important to improve the presentation of Sec. 4. In particular, I would appreciate a full worked example for CyclER and a better explanation of Fig 1 (the label is misleading: there is no actual formal MDP and the part on the very right isn’t really explained). It would be helpful to have the experiments available online to independently verify them.
>
> Thank you to the reviewer for their suggestions! We have modified section 4 to include a fully worked out example for CyclER that follows the running example from Figure 1. We have also changed section 2 to introduce Figure 1 earlier and clarify the FlatWorld MDP. Regarding the availability of the experiments: in the supplementary material, we have attached the codebase we used to run our experiments, along with instructions to reproduce the results. Upon deanonymization of the paper, we will attach an online link to the code repository for easy access and maintenance.
>
> > What is the intuition why re-distribution of rewards in CyclER help? I guess it depends on how advantage is estimated in PPO? Is bootstrapping used?
>
> Similar to other works on reward shaping [1][2], the redistribution of rewards that CyclER provides relieves reward sparsity that is otherwise present in existing LTL-guided RL approaches. Rather than having rewards for transitions that are almost entirely zeros and very few (if any) ones, CyclER allows for more transitions to have nonzero rewards. This means that in an on-policy update, the value (or advantage) estimates will improve in accurately guiding the policy towards task completion. CyclER is agnostic to how the advantage or value estimates are computed (so long as they are correctly computed) and can be used by any on-policy algorithm. There is no bootstrapping used or required by CyclER.
>
> [1] Reward Shaping in Episodic Reinforcement Learning. Grzes. AAMAS 2017.
>
> [2] Potential-Based Reward Shaping For Intrinsic Motivation. Forbes et. al. AAMAS 2024.
>
> > It is not clear how CyclER with QS can provide a dense reward signal. E.g. consider a reachability task which requires lots of transitions in the MDP but only self-loops in the initial state of the LDBA are taken. Then it seems to me that your method only provides small rewards twice: the first time the loop is taken and when the accepting state is reached. On the other hand, a large number of transitions in the MDP, which make slow but steady progress, remain unrewarded. Am I missing something?
>
> Your understanding of the example you have provided is correct for CyclER without QS. However, with QS, you get a dense reward exactly shaped by this “slow and steady progress”, provided that accurate robustness functions are given. Consider as an example the task specification $G(F(r))$ in our FlatWorld environment, but under slightly different dynamics: every time the red region is reached the agent gets teleported back to its initial location in the MDP. Under the non-QS version of CyclER defined in objective (6), a positive reward will be administered every time the red region is visited, but not otherwise. This reward structure is indeed identical to a reward structure that offers positive reward only when the accepting state of the LDBA is visited.
>
> Now suppose we use the QS version of CyclER. We first assign the propositional variable $r$ a robustness function. A reasonable choice (that we also use in our experiments) is the negative signed distance to the red region from the agent’s current location in the MDP. For $r$ to be evaluated as true, $-\text{distance}(r, s) <= 0$. Under objective (7), even if a transition in the MDP does not reach the red region, as long as it makes progress by getting closer to the red region, it will receive a positive reward. If an agent makes small, but progressive transitions towards the red region, it will get a positive nonzero reward for every single transition. In Appendix section E we describe this procedure in detail for the running example which provides the same intuition as the toy example we provide here.

---

> > ### Author Response · Authors · 2025-02-01
> > **Author response to reviewer tqsv (continued)**
> >
> > > Are there theoretical guarantees for CyclER with quantitative semantics?
> >
> > This is a good question - in its current form, CyclER with QS does not have any theoretical guarantees. The primary reason for this is that we allow for any arbitrary (valid) robustness functions to be used for the APs in the LTL formula. In not making any assumptions regarding these robustness functions, there are a number of degenerate cases that can arise that would invalidate the theoretical guarantee we make for (non-QS) CyclER in Theorem 4.1.
> >
> > To see this, consider an example from the FlatWorld environment in the paper where an agent has to first visit the red region, then the blue region in that order. The robustness functions we use for each goal region is the negative signed distance from the agent to any part of the region. In other words, the agent will receive a reward proportional to the distance it had to travel to go from its position in one region (or the initial location, to start) to the next region. So in this example, the agent first will receive a reward proportional to the distance from the initial location to the red region. Once it visits any part of the red region, it will then receive a reward proportional to distance between its location upon visiting the red region and the blue region.
> > As a result, the agent will be incentivized to visit the outer edge of the red region furthest from the blue region (the lower right corner), so that it can travel the furthest distance possible to get to the blue region from the red region. Suppose there was a pit of lava near the lower right corner of the red region that the agent could fall into with a very small probability. If the probability is low enough, the agent will still be incentivized to exhibit the aforementioned degenerate behavior because the increase in reward will outweigh the probability of failure.
> >
> > Now, some of these degenerate cases can be addressed by correctly setting the discounting factor, but the assumptions required for the discount factor, robustness functions, and MDP are highly complex. We see the study of theoretical guarantees in the QS setting as an exciting direction for future work. In the scope of our paper, we instead use CyclER-QS as a heuristic (as mentioned in sections 4 and 5). With reasonable choices of robustness functions, the approach works well, as evidenced by our experiments.
> >
> > > I think the paper might have benefitted from focussing on the simpler problem of RL where the single objective is to maximise LTL objectives. The constrained setting mostly adds orthogonal challenges (although reward sparsity becomes an even bigger problem). That way, the presentation could be improved and space is created to explain the main ideas CyclER with QS in sufficient detail in the main body.
> >
> > The original idea and motivation of CyclER came from studying the problem of LTL-constrained RL. We do agree that the challenge of reward sparsity is of independent interest from the constrained setting - accordingly, in our experiments, we compare against a baseline that does not act in the constrained setting (LCER-no-MDP) against CyclER in the constrained (section 6) and unconstrained (Appendix section D) setting. However, we see the overall story as our contribution: LTL-constrained reward optimization can soundly be defined as a simple linear combination of MDP reward and LTL reward proxy, under some assumptions (see section 3). In practice, CyclER addresses the challenges of this problem setting, namely reward sparsity. In any case, we agree with the reviewer on their suggestion to add detail and explanation regarding CyclER to the main text, and have done so in section 4.

---

> > > ### Author Response · Authors · 2025-02-01
> > > **Author response to reviewer tqsv (continued)**
> > >
> > > > The environments seem pretty simple and similar. There is no empirical comparison to other works for the RL + LTL special case apart from Voloshin et al. 2023.
> > >
> > > Yes, there are similarities in that all environments and tasks are variants of loco-navigation. In the Safety-Gym environments, the dynamics are complex in terms of dimensionality and both the size and continuity of state/action space, but it’s also true that the control problems themselves in this (and all) domains are pretty straightforward relative to other problems in control (e.g., dexterous manipulation, robot systems with high DoF). Our intention in our experimental evaluation is more concerned with task complexity and how CyclER compares to existing approaches in handling them, and the control aspect is a bit orthogonal. Our intention in selecting environments was to ensure that the difficulty of the control problem did not obscure the experimental takeaways we demonstrate.
> > >
> > > We also note that our experimental domains are both consistent with and often more complex than prior work on temporal logic-constrained policy optimization [1] [2], as well as the broader LTL-guided deep RL space [3] [4]. We see the scaling up of control complexity for CyclER-guided tasks as a compelling direction for future work.
> > >
> > > [1] Policy Optimization with Linear Temporal Logic Constraints. Voloshin et. al. NeurIPS 2022.
> > >
> > > [2] Deep reinforcement learning under signal temporal logic constraints using Lagrangian Relaxation. Ushio et. al. IEEE Access 2022.
> > >
> > > [3] Temporal-Logic-Based Reward Shaping for Continuing Reinforcement Learning Tasks. Jiang et. al. AAAI 2021.
> > >
> > > [4] Temporal Logic Specification-Conditioned Decision Transformer for Offline Safe Reinforcement Learning. Guo et. al. ICML 2024.
> > >
> > > Lastly, we thank the reviewer deeply for their careful review and suggestions on typographical errors and clarity issues! We have revised the text in an effort to fix and address these.
> > >
> > > We would like to sincerely thank the reviewer again for the suggestions and feedback, which we have incorporated into our updated manuscript. We feel that this has significantly strengthened the quality and clarity of our paper. If the reviewer has additional questions, we are happy to address them.

---

> > ### Comment · Reviewer_tqsv · 2025-02-03
> > **Further Clarifications**
> >
> > Thank you very much for the response. I would like to seek further clarification regarding the following two issues:
> >
> > 1. My understanding of the advantage estimation is that it uses full Monte-Carlo roll-outs and no bootstrapping. As an alternative to Algorithm 1, it would be conceivable to give reward 1 in accepting states and only give the partial credit you propose in the final unfinished cycle. Modulo discounting, the advantages should be roughly the same (given that full roll-outs are used). Can you comment?
> >
> > 2. How is $e$ updated for CyclER with QS? My understanding is that, as for CyclER without QS, $e$ is set to $0$ in accepting states and it is set to $1$ when a particular transition is taken. (Again, I think it would be helpful to add this explicitly to Algorithm 1.) In other words, a single transition can only receive a positive reward once before visiting an accepting state. I seem to be missing something as this contradicts your response and I would appreciate further clarification.

---

> > > ### Author Response · Authors · 2025-02-03
> > > **Follow-up response to reviewer tqsv**
> > >
> > > Thank you to the reviewer for their continued engagement with our work and for their helpful feedback! We address each point raised in turn:
> > >
> > > > My understanding of the advantage estimation is that it uses full Monte-Carlo roll-outs and no bootstrapping. As an alternative to Algorithm 1, it would be conceivable to give reward 1 in accepting states and only give the partial credit you propose in the final unfinished cycle. Modulo discounting, the advantages should be roughly the same (given that full roll-outs are used). Can you comment?
> > >
> > > Yes, you are correct in your understanding of the advantage estimation. We agree that the reward structure you propose can certainly be used as an alternative to Algorithm 1, and that the advantage estimates in principle should not greatly differ. As is often the case in deep RL implementations, however, these changes in structure can make significant performance differences. For example, at a point in our implementation, we experimented with maintaining a replay buffer of trajectories, and sampling from both 'good' trajectories (those that accumulated nonzero reward) and 'bad' trajectories previously collected during policy updates. One aspect of your proposed reward structure that may cause inconsistencies is that identical transitions in different trajectories may have different rewards. In a purely on-policy update scheme, this is *probably* fine, but this inconsistency could cause instabilities when moving to a learning scheme that shifts off-policy. However, there are benefits to the proposed structure as well. A thorough evaluation of the potentially many variants of CyclER would make for compelling future work, and we thank the reviewer for the insight.
> > >
> > > > How is $e$ updated for CyclER with QS? My understanding is that, as for CyclER without QS, $e$ is set to $0$ in accepting states and it is set to $1$ when a particular transition is taken. (Again, I think it would be helpful to add this explicitly to Algorithm 1.) In other words, a single transition can only receive a positive reward once before visiting an accepting state. I seem to be missing something as this contradicts your response and I would appreciate further clarification.
> > >
> > > We apologize if there seems to be any contradiction. Your understanding for CyclER without QS is correct. For CyclER with QS, the semantics are analogous and follow objective (7): before a particular transition is taken (i.e., $e$ is zero for that transition), nonzero rewards will be given to an agent as determined by (7). When a transition is taken, $e$ will be set to $1$, just as is done in the non-QS version. Now, no reward will be given for either making progress towards that transition within the cycle, or taking the transition itself. The intuition behind maintaining $e$ is that it prevents reward hacking; that is, an agent cannot "farm" reward from a single transition along the cycle and avoid visiting the accepting state. In doing so, we are disincentivizing an agent from progressing "backwards" in the LDBA and having to take a transition more than once. We provide this intuition via example in appendix section F. If you are pointing out that " a single transition can only receive a positive reward once before visiting an accepting state" is contradicting this response, that is true: this statement holds only for the non-QS version. To help clear up this understanding, we have adjusted the wording in section 4.3 (second to last paragraph), and have followed the reviewer's suggestion and added explicit bookkeeping of the visited frontier $e$ to Algorithm 1.
> > >
> > > We thank the reviewer again for their continued discussion and helpful suggestions. We have incorporated their feedback into our revised manuscript and do feel that the work has improved as a result. Please let us know if you have further questions and we are happy to further engage in discussion.

---

> > > > ### Comment · Reviewer_tqsv · 2025-02-04
> > > >
> > > > Thank you for the clarifications. I do not have further questions.

---

### Decision · Action_Editor_qdzj · 2025-03-05

**Recommendation:** Accept as is

**Comment:**

The reviewers initially pointed out several issues of clarity. However, by engaging in a continuing discussion with the reviewers and updating the manuscript accordingly, the authors have significantly improved the clarity of the work. They have also conducted additional experiments to better support their claims. At this point, I consider the paper ready for publication.

**Audience:**

This work is of interest for the RL community, especially those interested in safe RL or in combining symbolic tools with RL

**Claims And Evidence:**

The claims are supported by convincing empirical evidence.